# The QBO and global-scale tropical waves in Aeolus wind observations, radiosonde data, and reanalyses

Manfred Ern[1], Mohamadou A. Diallo[1], Dina Khordakova[1], Isabell Krisch[2], Peter Preusse[1], Oliver Reitebuch[2], Jörn Ungermann[1], and Martin Riese[1]

[1]Institut für Energie- und Klimaforschung – Stratosphäre (IEK–7), Forschungszentrum Jülich GmbH, Jülich, Germany
[2]Deutsches Zentrum für Luft- und Raumfahrt e.V. (DLR), Institut für Physik der Atmosphäre, Oberpfaffenhofen, Germany

**Correspondence:** M. Ern (m.ern@fz-juelich.de)

**Abstract.** The quasi-biennial oscillation (QBO) of the stratospheric tropical winds influences the global circulation over a wide range of latitudes and altitudes. Although it has strong effects on surface weather and climate, climate models have large difficulties in simulating a realistic QBO, especially in the lower stratosphere. Therefore, global wind observations in the tropical upper troposphere and lower stratosphere (UTLS) are of particular interest for investigating the QBO and the tropical waves that contribute significantly to its driving. In our work, we focus on the years 2018–2022 and investigate the QBO and different tropical wave modes in the UTLS region using global wind observations by the Aeolus satellite instrument, and three meteorological reanalyses (ERA-5, JRA-55, and MERRA-2). Further, we compare these data with observations of selected radiosonde stations. By comparison with Aeolus observations, we find that on zonal average the QBO in the lower stratosphere is well represented in all three reanalyses, with ERA-5 performing best. Averaged over the years 2018–2022, agreement between Aeolus and the reanalyses is better than 1 to $2\,\mathrm{m\,s^{-1}}$, with somewhat larger differences during some periods. Different from zonal averages, radiosonde stations provide only local observations and are therefore biased by global-scale tropical waves, which limits their use as a QBO standard. While reanalyses perform well on zonal average, there can be considerable local biases between reanalyses and radiosondes. We also find that, in the tropical UTLS, zonal wind variances of stationary waves and the most prominent global-scale traveling equatorial wave modes, such as Kelvin waves, Rossby-gravity waves, and equatorial Rossby waves, are in good agreement between Aeolus and all three reanalyses (in most cases better than 20% of the peak values in the UTLS). On zonal average, this supports the use of reanalyses as a reference for comparison with free-running climate models, while locally certain biases exist, particularly in the QBO wind shear zones, and around the 2019–2020 QBO disruption.

## 1 Introduction

In the tropical stratosphere, the zonal wind exhibits characteristic oscillations, the quasi-biennial oscillation (QBO) and the semiannual oscillation (SAO). The QBO has an average period of about $28\,\mathrm{months}$ and its amplitude maximum in the middle stratosphere, while the amplitude maximum of the SAO is located in the stratopause region. An overview is given, for example,

in Baldwin et al. (2001), or Anstey et al. (2022). These oscillations are major modes of climate variability that affect the global circulation and climate system over a large range of altitudes and latitudes.

For example, the QBO has effects on deep convection and precipitation in the tropics (e.g., Liess and Geller, 2012). In the extratropics, the QBO has an effect on the frequency of sudden stratospheric warmings (SSWs), i.e., the circulation in the extratropical stratosphere (e.g., Holton and Tan, 1980; Gray et al., 2020), and on surface weather and climate (e.g., Baldwin and Dunkerton, 2001; Marshall and Scaife, 2009; Kidston et al., 2015). An overview of the surface impacts of the QBO is given, for example, in Anstey and Shepherd (2014) and Gray et al. (2018).

It was first proposed by Lindzen and Holton (1968) and Holton and Lindzen (1972) that the QBO is driven by tropical waves. Later, it was shown that global-scale tropical wave modes and small scale gravity waves are contributing with about similar strength to the tropical momentum budget of the QBO (e.g., Dunkerton, 1997; Ern and Preusse, 2009a, b; Kawatani et al., 2010; Kim et al., 2013; Krismer and Giorgetta, 2014; Ern et al., 2014; Kim and Chun, 2015; Pahlavan et al., 2021a, b). Furthermore, the QBO winds are filtering the spectrum of upward propagating waves, and thereby the QBO has effect on the wave driving of

the SAO (e.g., Hamilton and Mahlmann, 1988; Ern et al., 2015, 2021). The waves that are driving the QBO are mostly generated in the troposphere with convection as the main wave generation process. This is the case for small scale gravity waves (e.g., Beres et al., 2004; Bushell et al., 2015; Kalisch et al., 2016; Trinh et al., 2016; Kang et al., 2018, and references therein), as well as for the characteristic global-scale equatorial waves modes (e.g., Salby and Garcia, 1987; Manzini and Hamilton, 1993; Bergman and Salby, 1994; Wheeler and Kiladis, 1999; Ricciardulli and Garcia, 2000; Ern et al., 2009b, and references therein).

The global-scale equatorial wave modes are trapped in the tropics, and can be either symmetric or antisymmetric with respect to the equator. A theory of these wave modes has been developed first by Matsuno (1966). One of the most prominent wave modes in zonal winds are equatorial Kelvin waves. In zonal winds, Kelvin waves are symmetric with respect to the equator. Another wave mode are mixed Rossby-gravity waves (MRGWs). The latter are antisymmetric in zonal winds. Further, there are equatorial Rossby (ER) wave modes, and inertia-gravity wave (IGW) modes. These can be either symmetric or antisymmetric.

An overview of the horizontal structure of the different wave modes is given, for example, in Yang et al. (2003). Because these wave modes occur in characteristic wave bands, they can be identified in zonal wavenumber-frequency power spectra. Global observations from satellite are particularly suited for these kind of investigations: An overview of the spectral contributions of the different wave modes in the troposphere is given, for example, by Wheeler and Kiladis (1999) based on outgoing longwave radiation (OLR) observations. In the stratosphere, the spectral contributions of the different wave modes were investigated, for

example, by Ern et al. (2008) using global temperature observations from space.

In spite of its importance, simulating a realistic QBO with free-running state-of-the-science climate models is very difficult, as was shown in recent model intercomparisons (Bushell et al., 2022; Richter et al., 2022). One of the main reasons is that the wave driving of the QBO is not well represented in the models. This includes both the driving by gravity waves (e.g., Richter et al., 2022), but also the driving by tropical global-scale wave modes (e.g., Holt et al., 2022). Because of their small

spatial scales gravity waves are usually not resolved in climate models. Therefore their effect on the global circulation has to be approximated using different parameterization schemes, which involves large uncertainties. However, there are also large uncertainties related to the tropical global-scale waves. In particular, it was found by Holt et al. (2022) that in free-running

climate models the tropical global-scale waves in the upper troposphere (where these waves are coupled to convection) are often not very realistic.

Tropical waves are not only important for the driving of the QBO. In the tropics, upwelling of air from the troposphere into the stratosphere strongly influences the global distributions of trace species, such as ozone and water vapor (e.g., Butchart, 2014; Diallo et al., 2022), and particularly the mixing ratios in the upper troposphere and lower stratosphere (UTLS) are important for the global radiation budget (e.g., Riese et al., 2012). It has been shown that the transport of trace species through the tropical tropopause region is modulated by the effect of atmospheric waves: One effect is the upwelling by the wave-

induced Brewer-Dobson circulation (e.g., Mote et al., 1996; Pumphrey et al., 2008; Butchart, 2014). Another effect, in the case of water vapor, is freeze-drying in wave-induced clouds (e.g., Fueglistaler et al., 2009; Dessler et al., 2014; Schoeberl et al., 2015; Dinh et al., 2016, and references therein). State-of-the-science climate models, however, have large difficulties to simulate realistic distributions of tropical waves, and large inter-model differences in the upwelling in the tropical tropopause layer (TTL) have been attributed to differences in simulated tropical planetary waves and midlatitude synoptic waves (e.g.,

Yoshida et al., 2018). Therefore, observations of tropical global-scale waves, particularly in the UTLS, are needed to provide a reference and guidance for climate models.

This shows the importance of observing tropical waves globally in the UTLS region. Previous global observations of tropical waves, however, are mostly based on temperatures (e.g., Randel and Wu, 2005; Venkat Ratnam et al., 2006; Alexander et al., 2008; Ern et al., 2008; Alexander and Ortland, 2010; Cairo et al., 2010), which is complicated by the sharp vertical temperature

structure of the tropical tropopause. In particular, the change in static stability (i.e., the buoyancy frequency $N$) at the tropopause can cause jumps in wave amplitude, which makes it difficult to track wave amplitudes across the tropopause in observations. Wind observations would be much better suited to investigate wave activity in the UTLS region. However, even the QBO was discovered relatively late (Ebdon, 1960; Reed et al., 1961; Angell and Korshover, 1964), because wind observations in the tropics are generally sparse. Even today, the winds observed at single radiosonde stations, are often used as a benchmark for

the QBO and for comparison with models (e.g., Naujokat, 1986).

With the launch of the Aeolus satellite in August 2018, this situation has changed. Aeolus provides global wind observations in the troposphere and lower stratosphere since September 2018. In our work, we compare Aeolus wind observations with different reanalyses. Reanalyses are the output from numerical weather prediction (NWP) models that are constrained by assimilation of observations. We consider the following reanalyses: the fifth generation of the European Centre for Medium-Range

Weather Forecasts (ECMWF) reanalysis (ERA-5), the Japanese 55-year Reanalysis (JRA-55) of the Japan Meteorological Agency (JMA), and the Modern-Era Retrospective Analysis for Research and Applications, Version 2 (MERRA-2) of the National Aeronautics and Space Administration (NASA). First, we want to find out how reliable the QBO is in these reanalyses. In addition, we compare Aeolus winds and reanalyses with wind observations at different radiosonde stations in the tropics. Further, Aeolus wind observations allow to investigate the wave activity of different global-scale tropical wave modes across

the tropopause, and to compare them with the reanalyses. In this way, we obtain information how suitable reanalyses are as a benchmark for free-running atmospheric models.

The data sets used (Aeolus wind observations, radiosonde data, and reanalyses) are introduced in Sect. 2. In Sect. 3, we investigate the mean zonal wind, in particular the QBO, near the equator in these datasets. After that, in Sect. 4, we compare the activity of global-scale tropical wave modes in Aeolus data and reanalyses. Finally, Sect. 5 gives a summary and conclusions.

## 2 Data sets used

### 2.1 The Aeolus instrument

The European Space Agency (ESA) satellite Aeolus (e.g., Stoffelen et al., 2005, 2020; Reitebuch, 2012; Reitebuch et al., 2020) was launched in August 2018 and provided Level 2B scientific wind products until 30 April 2023. Aeolus carries the Atmospheric LAser Doppler Instrument (ALADIN), which is the first wind lidar in space. ALADIN is a Doppler lidar operating at 354.8 nm wavelength. The lidar beam penetrates the atmosphere from above, and from the Doppler shift of the backscattered light the atmospheric wind speed parallel to the the line of sight (LOS) of the instrument is derived. Winds can be derived either from Mie scattering on aerosol and cloud particles, or from Rayleigh scattering on molecules. Here, we focus on the winds derived from Rayleigh backscattering because of the more regular global coverage compared to Aeolus Mie winds from Aerosol and cloud layers. The maximum altitude range of Aeolus Rayleigh winds is from the ground to about 30 km, but varies strongly depending on the commanded altitude setting. At low altitudes, Aeolus Rayleigh winds are also limited by opaque clouds.

Aeolus is in a dusk-dawn orbit with an inclination of $\sim 97°$ (ESA, 2023a) and, thus, provides near-global coverage. ALADIN is pointing downward with an off-nadir angle of $\sim 35°$, perpendicular to the flight direction. Typical horizontal and vertical averaging lengths are about 90 km and 0.5 to 2 km, respectively (e.g., Krisch et al., 2022, and references therein). Usually, effects of vertical winds are too weak to be seen in the Aeolus LOS winds, given the typical precision of Aeolus winds of about 3 to 7 m s$^{-1}$ (e.g., Rennie et al., 2021). A discussion of different error sources is given in Appendix B, including precision (Appendix B1) and potential biases due to vertical winds (Appendix B2).

Aeolus wind products are of high quality, as has been shown by several validations using airborne wind observations (e.g., Lux et al., 2020, 2022; Witschas et al., 2022), super pressure balloons (e.g., Bley et al., 2022), ground based stations (e.g., Iwai et al., 2021; Baars et al., 2022; Ratynski et al., 2023), or radiosondes (e.g., Baars et al., 2020; Iwai et al., 2021; Ratynski et al., 2023). Consequently, numerical weather predictions benefit from the assimilation of Aeolus winds (e.g., Rennie et al., 2021, 2022; Zagar et al., 2021; Garrett et al., 2022; Martin et al., 2023; Pourret et al., 2022; Feng and Pu, 2023), and first scientific studies were carried out (e.g., Banyard et al., 2021, 2022; Wright et al., 2021)

Due to the high inclination orbit of Aeolus, in the tropics the Aeolus horizontal line of sight (HLOS) deviates from the zonal direction by only about $10°$: the azimuth angle $\Phi'$ of the HLOS (measured clockwise from due north) is about $260°$ for ascending orbits, and about $100°$ for descending orbits (assuming that the HLOS points towards the satellite). If $(u, v, w)$ is the vector of atmospheric zonal, meridional, and vertical wind, neglecting the vertical wind $w$, the HLOS wind observed by

Aeolus is

$$HLOS_{wind} = -u\sin(\Phi') - v\cos(\Phi').$$ (1)

For an illustration of the viewing geometry see also https://confluence.ecmwf.int/display/AEOL/Aeolus+Level-2B+BUFR+ FAQ (last access: 03 March 2023) and https://confluence.ecmwf.int/download/attachments/46596815/aeolus_obs_operator. pdf?version=1&modificationDate=1524579985341&api=v2 (last access: 03 March 2023). Accordingly, the sensitivity of Aeolus $HLOS_{wind}$ to the zonal wind component relative to meridional wind component is

$$S_{uv} = \tan(\Phi').$$ (2)

This means that in the tropics Aeolus $HLOS_{wind}$ is about 6 times more sensitive to zonal wind than to meridional wind. Given this difference in sensitivity, and given that meridional wind is usually weaker than zonal wind, we assume that in the tropics Aeolus $HLOS_{wind}$ is exclusively caused by zonal winds, and we calculate zonal wind from Aeolus $HLOS_{wind}$ as

$$u = -HLOS_{wind}/\sin(\Phi').$$ (3)

This conversion is performed separately for each altitude in each altitude profile after interpolation to a set of fixed altitudes (see below). It was shown by Krisch et al. (2022) that over a large latitude range in the tropics and subtropics this is a very good approximation. For a detailed discussion of the Aeolus observing geometry see also Krisch et al. (2022). Further, potential effects caused by neglecting the meridional wind in the tropics are discussed in Appendix A2 and Appendix B3.

Our work is based on Aeolus level 2B HLOS winds using the data product versions 2B10, 2B11, 2B12, 2B13, 2B14, and 2B15, which, in combination, cover the whole period of Aeolus observations starting from 3 September 2018. Because Aeolus uses a terrain-following altitude setting, we interpolate Aeolus HLOS winds on a fixed vertical grid with $0.25\,\mathrm{km}$ vertical resolution, separately for each altitude profile (i.e., no horizontal interpolation is applied).

## 2.2 The IGRA radiosonde archive

We also make use of the Integrated Global Radiosonde Archive (IGRA) provided by the National Centers for Environmental Information (NCEI) of the National Oceanic and Atmospheric Administration (NOAA). This archive is a collection of historical and near real-time radiosonde and pilot balloon soundings around the globe, which is also used as input for reanalyses, for example the JRA-55 reanalysis (e.g., Durre et al., 2018). A description of IGRA and the related data processing is given in Durre and Yin (2008) and Durre et al. (2006, 2008, 2018).

Because Aeolus data are given on an altitude grid, we interpolate the radiosonde data on a fixed set of geopotential altitudes and use the same altitudes as for Aeolus. The calculation of geopotential height from radiosondes is described in Durre et al. (2008) (their Eq. (A1)). For this, it is assumed that the hydrostatic balance applies, and the difference $dZ$ in geopotential height between two levels $i$ and $i+1$ is given by:

$$dZ = \frac{R}{g}\frac{T_i + T_{i+1}}{2}\ln\frac{p_i}{p_{i+1}}$$ (4)

where $T_i$ and $T_{i+1}$ are the atmospheric temperatures, and $p_i$ and $p_{i+1}$ are the atmospheric pressures at the levels $i$ and $i+1$, respectively, $R$ is the ideal gas constant, and $g$ is Earth's gravity acceleration. This means that geopotential height can be obtained in an iterative way based on the observed temperature-pressure profile.

## 2.3  The reanalyses ERA-5, JRA-55, and MERRA-2

In our study, we use 6-hourly data of three different reanalyses that are interpolated to a $1°$ by $1°$ latitude-longitude grid. Further, we interpolate the reanalysis data from the model levels to a set of fixed geopotential altitudes with a vertical step of $0.5\,\mathrm{km}$. This resolution is coarser than for Aeolus and the radiosondes. Therefore, whenever differences between reanalyses and the other datasets are calculated, this is performed only for the altitude levels that are common to both datasets considered, i.e., with the coarser resolution of $0.5\,\mathrm{km}$ that is used for the reanalyses.

The European Centre for Medium-Range Weather Forecasts (ECMWF) fifth generation atmospheric reanalysis (ERA-5) (e.g., Hersbach et al., 2018, 2020) provides 137 levels in the vertical, and a high model top at $0.01\,\mathrm{hPa}$ ($\sim80\,\mathrm{km}$). Its vertical resolution is about 350 to $500\,\mathrm{m}$ in the tropical UTLS. Its horizontal resolution is TL639, corresponding to about $31\,\mathrm{km}$ at low latitudes, and it uses parameterizations for the effect of non-resolved orographic (Lott and Miller, 1997; Sandu et al., 2013) and nonorographic (Orr et al., 2010) gravity waves. Due to its high number of vertical levels, ERA-5 has a better vertical resolution than the other two reanalyses considered here, and also the horizontal resolution is considerably better.

The Japanese 55-year Reanalysis (JRA-55) (Kobayashi et al., 2015) of the Japanese Meteorological Agency (JMA) has 60 model levels with the model top at $0.1\,\mathrm{hPa}$, and a horizontal resolution of TL319 ($\sim55\,\mathrm{km}$ at low latitudes). Its vertical resolution is about 1 to $1.5\,\mathrm{km}$ in the tropical UTLS. It uses a parameterization for the effect of orographic gravity waves (Iwasaki et al., 1989a, b), but has no nonorographic gravity wave parameterization. Instead, Rayleigh damping is applied at pressures less than $50\,\mathrm{hPa}$ (altitudes above $\sim21\,\mathrm{km}$ in the tropics) to simulate the effect of small-scale gravity waves on the mean flow (see also Holton, 1982). In addition, the horizontal diffusion coefficient is gradually increased with altitude at pressures lower than $100\,\mathrm{hPa}$.

The Modern-Era Retrospective Analysis for Research and Applications, Version 2 (MERRA-2) reanalysis (Gelaro et al., 2017) is produced by the Global Modeling and Assimilation Office (GMAO) at the National Aeronautics and Space Administration (NASA) and uses the Goddard Earth Observing System (GEOS) model. MERRA-2 has 72 layers in the vertical with a model top at $0.01\,\mathrm{hPa}$, and a top layer mid level at $0.015\,\mathrm{hPa}$ ($\sim78\,\mathrm{km}$) in the upper mesosphere. Its vertical resolution is about $1.2\,\mathrm{km}$ in the tropical UTLS. The horizontal resolution is $0.5°$ latitude $\times$ $0.625°$ longitude. Parameterizations for both orographic (McFarlane, 1987) and nonorographic gravity waves (Garcia and Boville, 1994; Molod et al., 2015) are included. In particular, the MERRA-2 nonorographic gravity wave drag scheme was optimized for a better representation of the QBO and the SAO in the tropics (Molod et al., 2015). Additional damping is applied only at relatively high altitudes (at pressures less than $0.24\,\mathrm{hPa}$, $\sim58\,\mathrm{km}$).

All three reanalyses assimilate a large number of observations and are therefore much closer to reality than free-running atmospheric models. This means, they can be used as a guidance and reference for free-running models. Still, it is known that atmospheric models in general (including reanalyses) have problems particularly in the tropics, and an improved assimilation

of equatorially trapped waves using, for example, wind observations in the tropics, was suggested (e.g., Baker et al., 2014; Zagar et al., 2016, 2021). Validation of reanalysis data with wind observations in the tropics is therefore required. This will be performed in the following.

It should also be mentioned that ECMWF has started to assimilate Aeolus data into its operational weather forecast on 9 January 2020 (see: https://www.ecmwf.int/en/about/media-centre/news/2020/ecmwf-starts-assimilating-aeolus-wind-data; last access 03 March 2023), however, all reanalyses used in this study are independent and do not assimilate Aeolus data. In particular, even for ERA-5 this is not the case (see: https://confluence.ecmwf.int/display/CKB/ERA5%3A+data+documentation# ERA5:datadocumentation-Observations; last access 03 March 2023).

## 3   The QBO in Aeolus, radiosonde and reanalysis data

### 3.1   A comparison of Aeolus and reanalysis equatorial zonal average zonal wind

In a first step, we address zonal average zonal winds near the equator. For this purpose, we average Aeolus zonal winds over the latitude band 2°S–2°N. Further, we average Aeolus zonal winds over overlapping time intervals of $7\,\mathrm{days}$ with a time step of $3\,\mathrm{days}$. In this way, the effect of measurement noise and of small-scale gravity waves is much reduced. The same procedure is applied to the zonal winds from the reanalyses.

The result is shown in Fig. 1a, d, g, and j for ERA-5, JRA-55, MERRA-2, and Aeolus respectively. For Aeolus (Fig. 1j), the altitude coverage changes with time, depending on measurement mode. One particular measurement mode, which was dedicated to observing the QBO and covered altitudes up to about $25\,\mathrm{km}$ in the tropics, was introduced on 17 June 2020 (ESA, 2023b). This QBO-mode was run every 7 days for always one day only, which means that, while this QBO-mode was performed, the Aeolus data coverage above $\sim 20\,\mathrm{km}$ is much reduced compared to other observation periods that continuously cover this altitude range. The QBO-mode was run until 19 January 2022. Starting from 24 January 2022 the Aeolus altitude range in the tropics was generally extended to $\sim 25\,\mathrm{km}$, which means a much better data coverage above $\sim 20\,\mathrm{km}$. The reason for this extension of the altitude range was the eruption of the Hunga Tonga–Hunga Ha'apai volcano on 21 January 2021 (see also Wright et al., 2022; Ern et al., 2022; Legras et al., 2022) and the intent to monitor the volcanic cloud.

As can be seen from Figs 1b, e, and h, there is very good agreement between the reanalyses and Aeolus. Generally, the agreement between Aeolus and all reanalyses is better than about $5\,\mathrm{m\,s^{-1}}$ for the 7-day average zonal averages. Above $25\,\mathrm{km}$, however, i.e. at altitudes higher than measured by Aeolus, deviations between the reanalyses can become larger. This can be seen, for example, in the second half of the year 2018 above $\sim 25\,\mathrm{km}$ in the descending QBO eastward phase when ERA-5 and JRA-55 winds are considerably stronger than MERRA-2 winds.

The agreement between Aeolus and the reanalyses is further illustrated in Fig. 1c, f, and i by showing the differences between Aeolus and the respective reanalysis, averaged over the period 2018–2022 (red lines in Fig. 1c, f, and i). As can be seen from Fig. 1c, f, and i, ERA-5 shows the weakest biases ($< 1\,\mathrm{m\,s^{-1}}$) with respect to Aeolus, followed by MERRA-2 ($< 1.5\,\mathrm{m\,s^{-1}}$), and JRA-55 ($< 2\,\mathrm{m\,s^{-1}}$). Also given in Fig. 1c, f, and i as blue envelopes are the standard deviations of the differences between

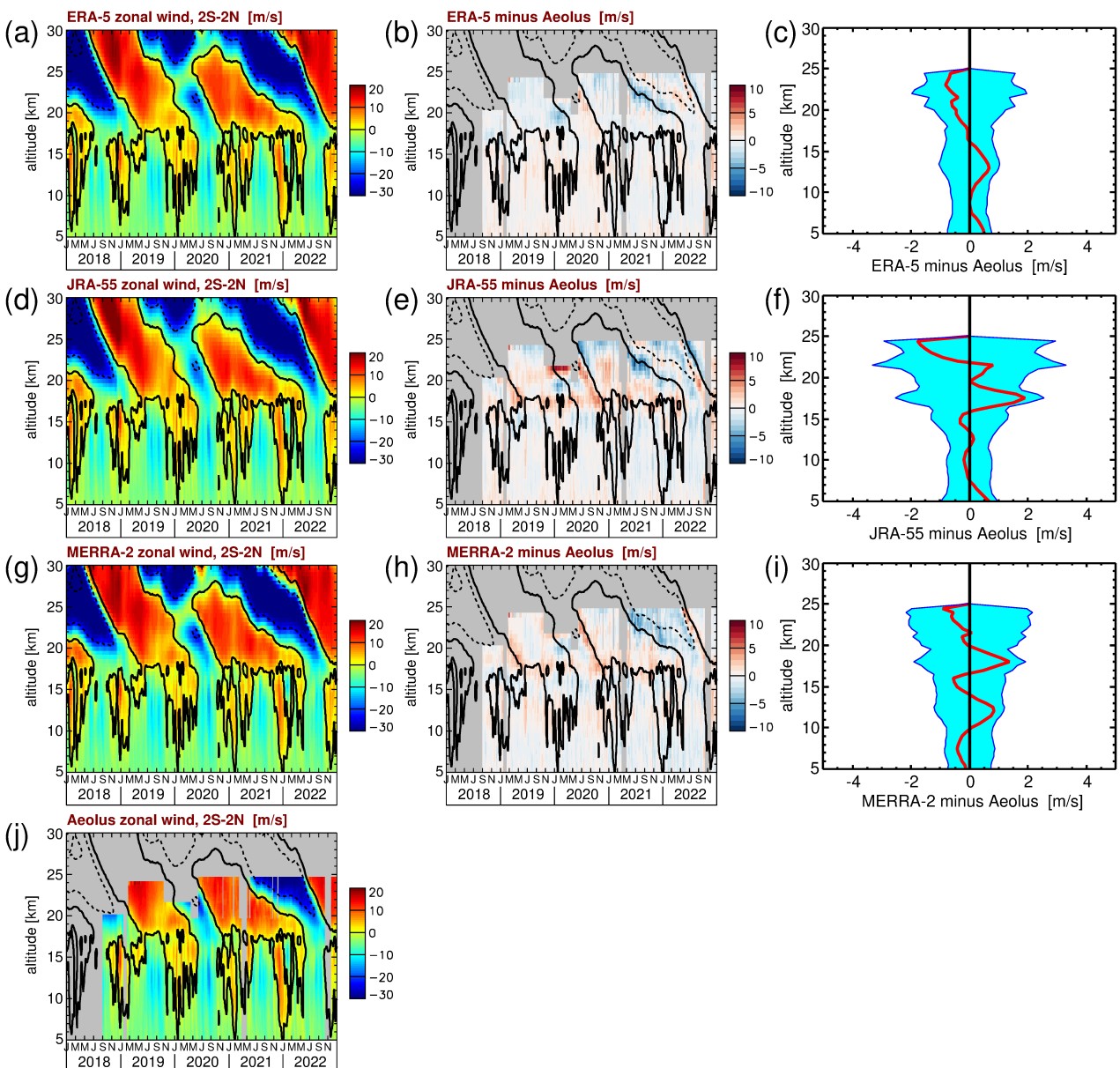

**Figure 1.** Altitude-time cross section of 2°S–2°N zonal mean zonal wind for **(a)** ERA-5, **(d)** JRA-55, **(g)** MERRA-2, and **(j)** Aeolus. Also shown are altitude-time cross sections of the differences **(b)** ERA-5 minus Aeolus, **(e)** JRA-55 minus Aeolus, and **(h)** MERRA-2 minus Aeolus, as well as the standard deviations of the differences to Aeolus for the period 2018–2022 (blue shaded envelope) and the 2018–2022 mean differences (red line) between Aeolus and **(c)** ERA-5, **(f)** JRA-55, and **(i)** MERRA-2. Contour lines in (a), (b), (d), (e), (g), (h), and (j) indicate the ERA-5 zonal mean zonal wind shown in (a). Contour lines are plotted every $20\,\mathrm{m\,s^{-1}}$. Dashed (solid) contour lines indicate westward (zero) wind.

Aeolus and the respective reanalysis. As we use Aeolus as a reference, i.e., the line of zero deviation in Fig. 1c, f, and i, the
blue envelopes are plotted around this zero line.

Particularly for ERA-5 and MERRA-2 the standard deviations in the altitude range 5 to $24\,\mathrm{km}$ are below $\sim 2\,\mathrm{m\,s^{-1}}$. Only
for JRA-55 the standard deviation exceeds $2\,\mathrm{m\,s^{-1}}$ in a few altitude ranges where biases between Aeolus and JRA-55 are
somewhat stronger. In particular, there is a strong positive deviation for JRA-55 from November 2019 until April 2020 at
altitudes above $\sim 21\,\mathrm{km}$ that seems to be related to the 2019–2020 QBO disruption, while many other deviations seem to be
related to wind shear zones.

Overall, this underlines the very good agreement between Aeolus and the three reanalyses considered here, which means
that the zonal average zonal wind from the reanalyses can be used as a reference for climate models. It should, however, be
kept in mind that differences between the reanalyses are getting stronger above $25\,\mathrm{km}$.

### 3.2 Investigation of local effects by comparison with radiosonde zonal wind

Because in zonal averages local effects will average out, we will next compare Aeolus and reanalysis zonal winds with obser-
vations at several radiosonde stations in the tropics. We selected eight stations in the approximate latitude band 5°S–5°N. We
took care that the locations of these stations cover a relatively wide range of longitudes. Further, observations at each station
should provide a good altitude and time coverage. The selected stations and their locations are shown in Fig. 2.

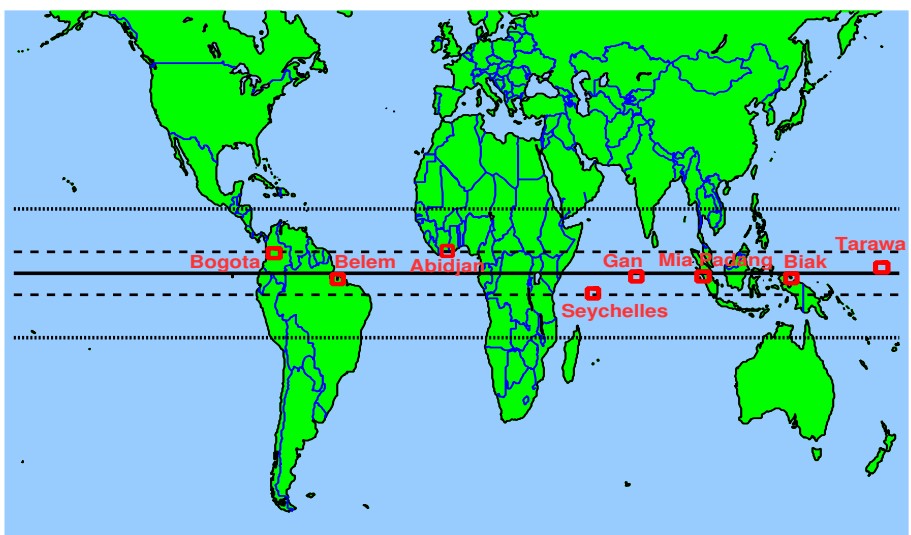

**Figure 2.** Locations of selected radiosonde stations in the tropics. The bold horizontal line marks the equator. The dashed horizontal lines
indicate the latitudes of 5°S and 5°N, and the dotted horizontal lines indicate the latitudes 15°S and 15°N.

Figure 3 shows the zonal wind observed at the eight radiosonde stations introduced in Fig. 2. Like in Sect. 3.1, we average
the wind observations in the same overlapping 7-day time intervals. Again, our purpose is to reduce the effect of measurement

noise and of observed atmospheric fluctuations due to small-scale gravity waves. Reducing the effect of gravity waves is important because we do not aim at achieving a perfect coincidence of observations in space and time.

There are several important findings from Fig. 3. Firstly, the zonal wind in the altitude range 10–15 km is prevalently positive (eastward) in approximately the Western Hemisphere (i.e., at Bogota and Belem), and prevalently negative (westward) in approximately the Eastern Hemisphere (i.e., at the Seychelles, Gan, Mia Padang, and Biak). This indicates the presence of a very strong quasi-stationary zonal wavenumber 1 wave of the zonal wind in this altitude range. Peak differences between different stations in this altitude range are as strong as $\sim 40\,\mathrm{m\,s^{-1}}$, which hints at a wavenumber 1 amplitude of sometimes as strong as $20\,\mathrm{m\,s^{-1}}$, or more. Above $\sim 15\,\mathrm{km}$ we also observe at all stations narrow bands of zonal wind fluctuations that propagate downward with time. As discussed, for example, in Cairo et al. (2010), this indicates the presence of upward-propagating traveling waves.

Since wind observations in the tropics are sparse, the zonal winds of single radiosonde stations are often used as a standard for the QBO (e.g., Naujokat, 1986). The limitations of this approach will be investigated in the following. Figure 4 shows the zonal wind differences between all other radiosonde stations introduced in Fig. 2 and Mia Padang as a reference. Mia Padang is selected as a reference because it is relatively close to the equator and has a good temporal coverage.

Of course, there are large zonal wind differences in the troposphere that are related to the quasi-stationary wave pattern that was mentioned above. Further, we can see from Fig. 4 that in the stratosphere for the three stations that are located about 5° off-equator (Bogota, Abidjan, and the Seychelles) there are persistent zonal wind differences of up to about $10\,\mathrm{m\,s^{-1}}$ over several months. This means that the zonal wind observed at these stations would not be a good QBO proxy, and stations closer to the equator would be preferable.

However, in the stratosphere there are also considerable zonal wind differences between Mia Padang and the other stations that are located close to the equator. Partly, these differences are caused by traveling global-scale waves. This effect is most strongly seen in the differences between Belem and Mia Padang — probably because the difference in longitude between these two stations is relatively large such that zonal wavenumber 1 waves are almost in anti-phase at these two stations. As most of the traveling waves seem to have periods of 30 days, or shorter, the effect of these waves will be strongly reduced in monthly averages.

Interestingly, there are also differences between Belem and Mia Padang that persist for several months. For example, in 2019 between 18 and 25 km altitude there are differences of up to $-10\,\mathrm{m\,s^{-1}}$ that persist for about three months. Similar negative deviations, albeit weaker, are also seen at Tarawa in the same months and the same altitude range. These deviations may therefore be related to a quasi-stationary wave. Deviations of similar magnitude and duration, but opposite sign are seen between Tarawa and Mia Padang in early 2019 at 20 to 30 km altitude, and in late 2020 to early 2021 at 18 to 23 km altitude. These differences occur even though the three mentioned stations are all closer than 1.5° to the equator. This means that observations at single stations can easily be biased and this shows the importance of global wind observations from satellite, such as Aeolus, in order to be able to calculate reliable zonal mean zonal winds in the stratosphere.

Because there are large zonal wind differences between the local radiosonde stations, this raises the question whether re-analyses are capable to reliably reproduce the zonal winds at the different locations. For a direct comparison with the local

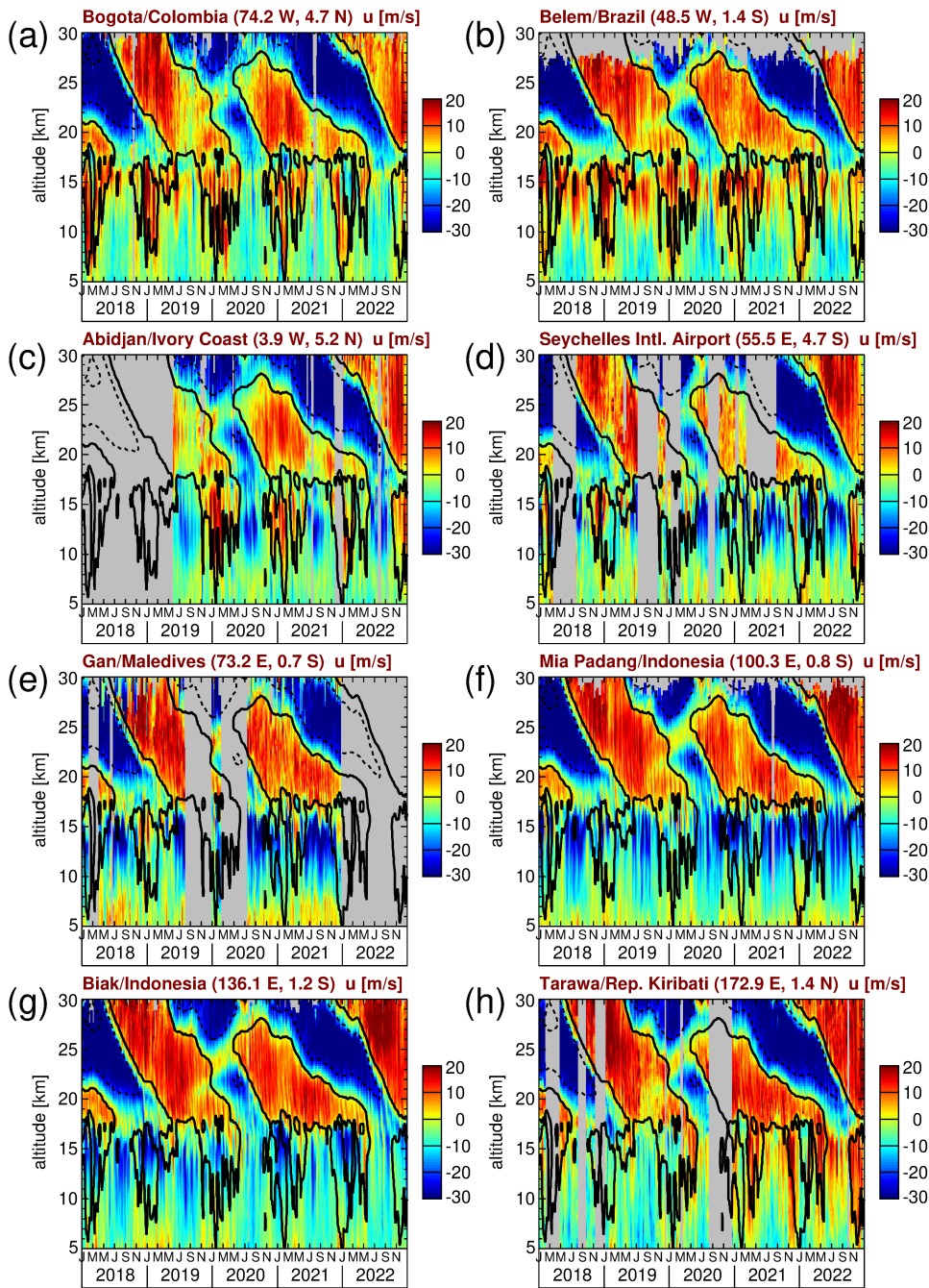

**Figure 3.** Zonal wind observed at the eight radiosonde stations introduced in Fig. 2. Contour lines indicate the ERA-5 zonal mean zonal wind shown in Fig 1a. Contour lines are plotted every $20\,\mathrm{m\,s^{-1}}$. Dashed (solid) contour lines indicate westward (zero) wind.

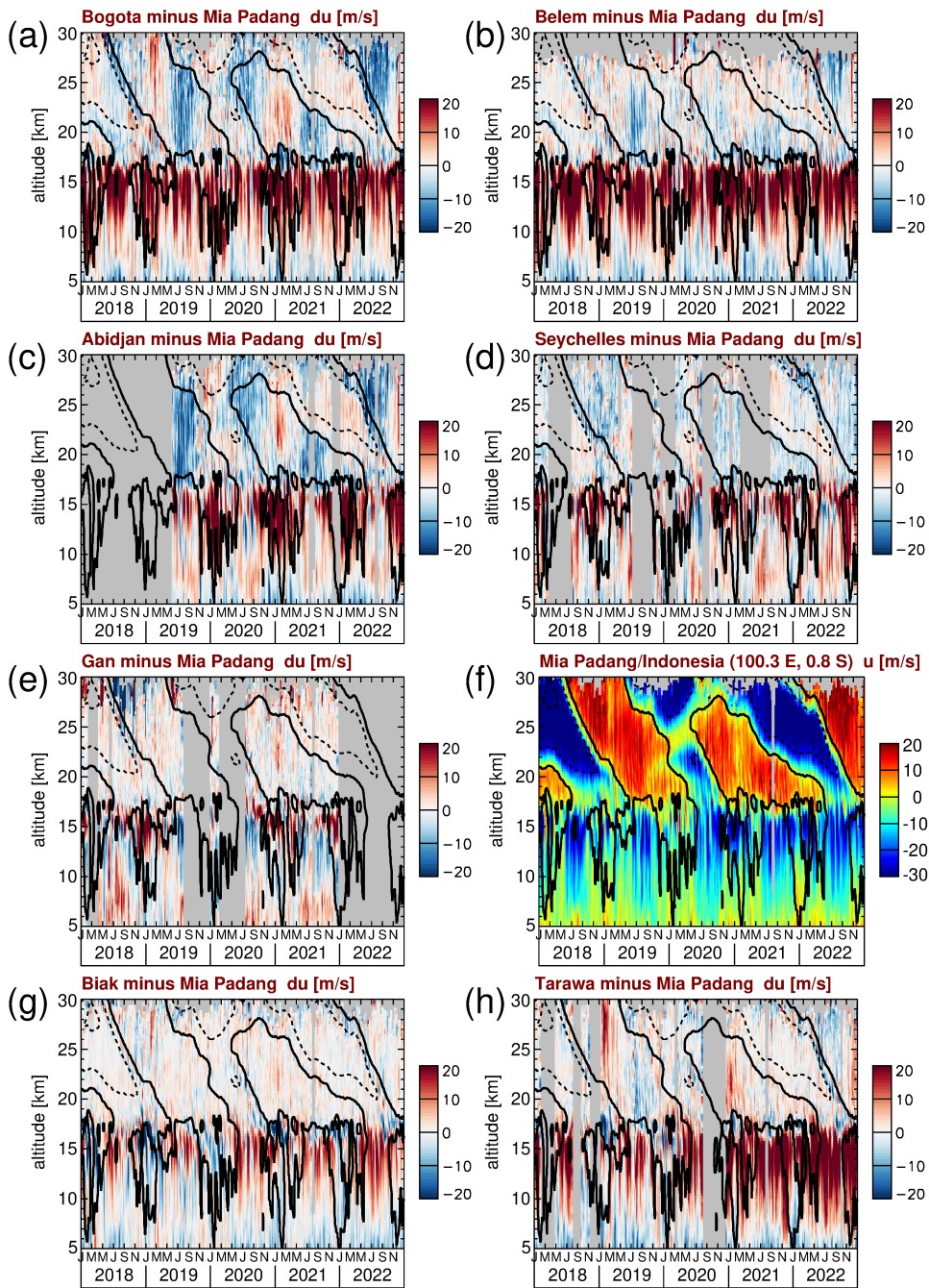

**Figure 4. (a–e)** and **(g–h)**: Zonal wind differences between seven of the eight radiosonde stations introduced in Fig. 2 and the winds at Mia Padang as a reference. In panel **(f)** the zonal winds at Mia Padang are also given for illustration. Contour lines indicate the ERA-5 zonal mean zonal wind shown in Fig 1a. Contour lines are plotted every $20\,\mathrm{m\,s^{-1}}$. Dashed (solid) contour lines indicate westward (zero) wind.

stations, we create time series at the different locations for the reanalyses and Aeolus observations. For the reanalyses, we average the zonal wind of the four grid points surrounding the respective location of the radiosonde station. As we use a $1° \times 1°$ latitude-longitude grid, this will introduce only little uncertainty due to spatial wind variations. For Aeolus, we average the Aeolus observations over an area of $\pm 2°$ latitude and $\pm 10°$ longitude, centered at the location of the radiosonde station. Also this larger averaging area will not introduce much uncertainty due to spatial variations of the zonal wind because we are only interested in large-scale variations. Again, we are averaging the data over 7 days and use the same set of 7-day averaging periods as mentioned above.

For the local comparisons, we select the two radiosonde stations Belem/Brazil and Mia Padang/Indonesia that are located in regions (South America, and the Maritime Continent, respectively) where numerous radiosonde stations are located that can be assimilated in the reanalyses. At these two stations, during most of the time, two radiosondes are launched per day. In addition, with the Seychelles and Tarawa/Kiribati, we select two stations that are located in the open ocean (the Indian Ocean, and the Pacific Ocean, respectively), where just a few radiosonde stations are located, which means that the input to the reanalysis data assimilation systems is relatively sparse. At the Seychelles and Tarawa usually only one radiosonde per day is launched. The differences between the radiosonde observations on the one hand, and Aeolus, or the reanalyses, on the other hand, are shown in Fig. 5 for Belem and the Seychelles, and in Fig. 6 for Mia Padang and Tarawa.

As expected, in Figs. 5 and 6 the differences between the local stations, on the one hand, and Aeolus and the reanalyses, on the other hand, are much larger than the differences on zonal average that were shown in Fig. 1b, e, and h. For Aeolus, we usually find only moderate differences to the radiosondes. Partly, these occur in zones of strong wind shear, and where the vertical extent of the QBO easterlies or westerlies is relatively narrow. These differences may be caused by effects of different altitude resolutions. In particular, for Aeolus the vertical resolution often is only 1.5 to 2 km at the highest altitudes. Further, there could be slight shifts between the altitude scales of Aeolus and the radiosondes. In addition, we find a strong scatter above $\sim 20$ km when the special Aeolus QBO observing mode with only one day per week of observations at high altitudes is performed (in the period from 17 June 2020 until 19 January 2022). Due to the data coverage of only one day per week, obviously, the effect of measurement noise and of small-scale gravity waves cannot be averaged out.

Also for the reanalyses we often find differences to the radiosondes in zones of strong vertical shear of the zonal wind. These shear zones are related to the QBO in the stratosphere, as well as to the tropopause region at 16 to 19 km altitude where strong vertical gradients of the zonal wind are related to the amplitude reduction with altitude of the strong quasi-stationary wave identified in Fig. 3. (Please note that the quasi-stationary wave cancels out on zonal average and is therefore not seen in the contour lines shown in Figs. 5 and 6.) Again, the zonal wind differences could be partly an effect of altitude resolution, but also the calculation of geopotential height (i.e., differences in the temperature-pressure altitude profiles) could introduce small differences in the altitude scales that, in the strong shear zones, can easily lead to certain differences in the zonal wind. A discussion of the effect of different altitude scales is given in Appendix B4. For the reanalyses another reason leading to the differences in the shear zones could be deficiencies in the zonal momentum budget that can lead to wind biases.

The shear zone related differences seen in Figs. 5 and 6 often consist of a pattern of alternating positive and negative differences, and are strongest for JRA-55, but are also seen for ERA-5 and MERRA-2. It should be mentioned that also

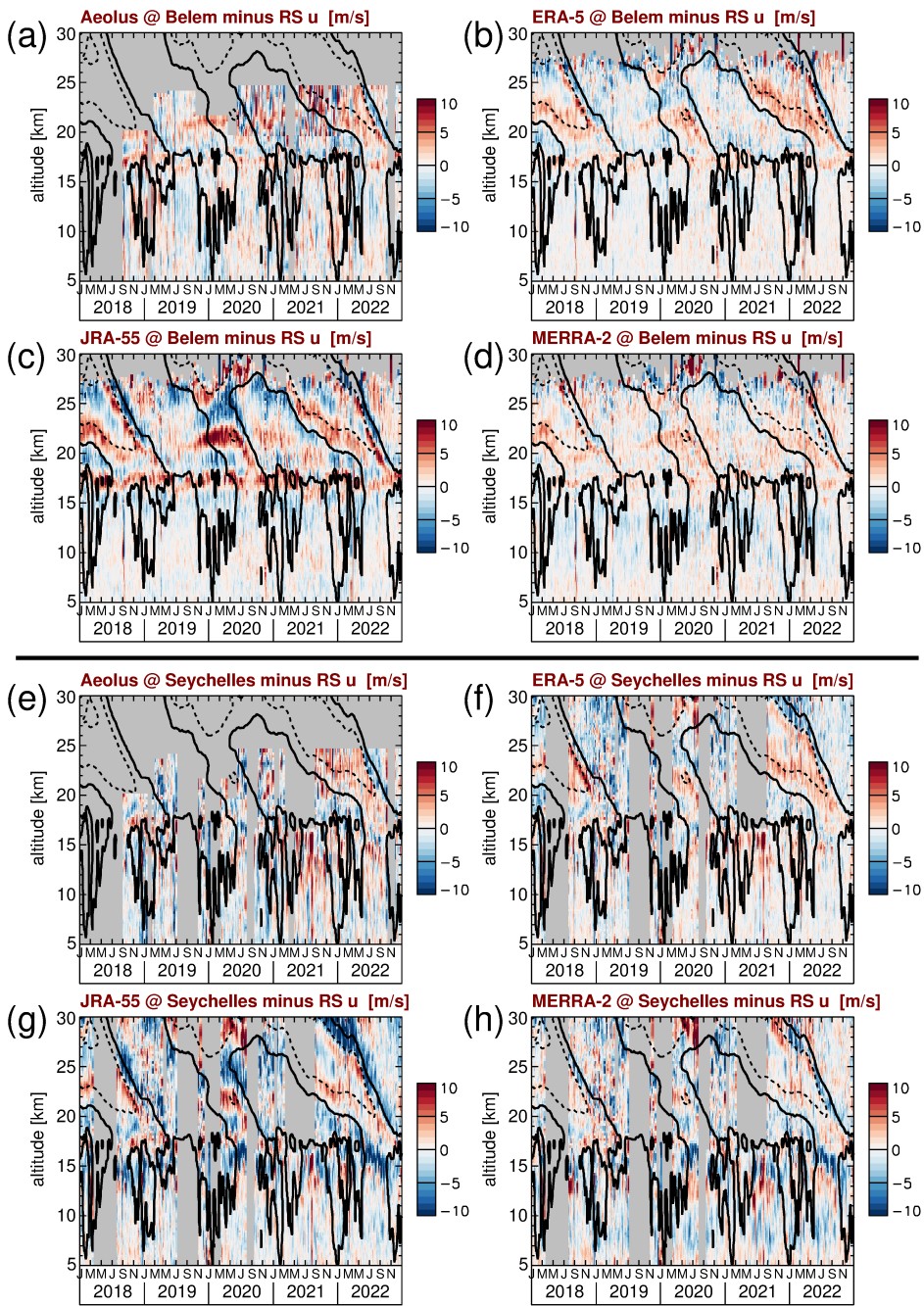

**Figure 5.** Zonal wind differences between the radiosonde (RS) observations at **(a)–(d)** Belem and Aeolus and the different reanalyses, as well as **(e)–(h)** Seychelles and Aeolus and the different reanalyses, Contour lines indicate the ERA-5 zonal mean zonal wind shown in Fig 1a. Contour lines are plotted every $20\,\mathrm{m\,s^{-1}}$. Dashed (solid) contour lines indicate westward (zero) wind.

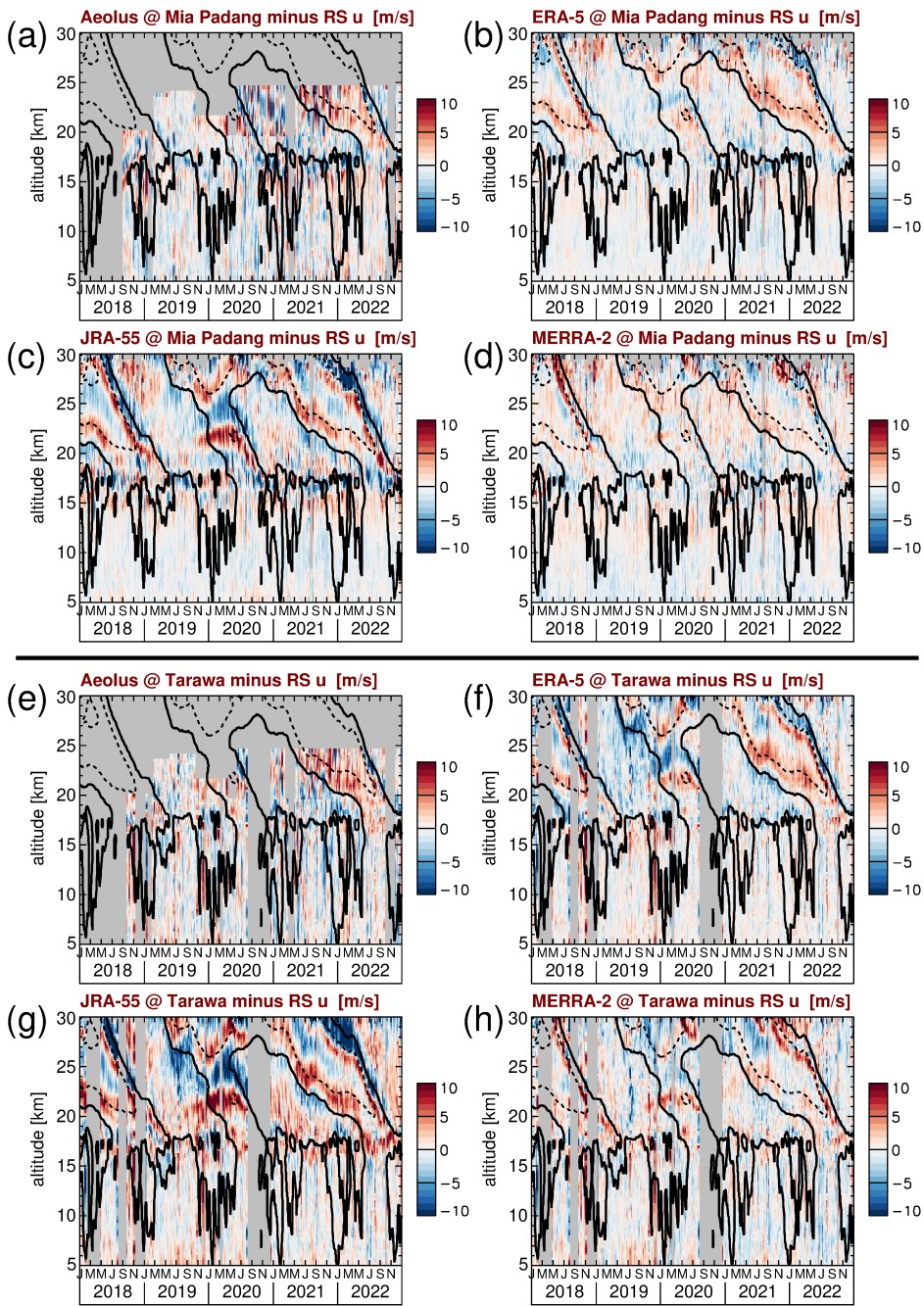

**Figure 6.** Zonal wind differences between the radiosonde (RS) observations at **(a)–(d)** Mia Padang and Aeolus and the different reanalyses, as well as **(e)–(h)** Tarawa and Aeolus and the different reanalyses, Contour lines indicate the ERA-5 zonal mean zonal wind shown in Fig 1a. Contour lines are plotted every $20\,\mathrm{m\,s^{-1}}$. Dashed (solid) contour lines indicate westward (zero) wind.

superpressure balloon observations in the tropics revealed stronger uncertainties of reanalysis winds in the QBO shear zones (e.g., Podglajen et al., 2014). Further, Podglajen et al. (2014) reported larger uncertainties of reanalysis datasets in regions of low radiosonde station density (the Indian Ocean and the Pacific Ocean), which is also seen in Figs. 5 and 6.

The fact that differences in the shear zones are less pronounced in the zonal averages shown in Fig. 1 could partly be related to cancellation effects that occur in zonal averages. Further, in the shear zones the differences between Aeolus and the radiosondes are often similar to the differences between the reanalyses and the radiosondes, which will reduce the shear zone effect when Aeolus is taken as a reference in Fig. 1. Finally, the altitude coverage of Aeolus at altitudes above 20 km is quite limited, which means that the altitudes where these effects are strongest are not well covered in Fig. 1.

Another effect that becomes evident from Figs. 5 and 6 is a pattern of differences between reanalyses and radiosondes that occurs from late 2019 to mid 2020 in the approximate altitude range 20–25 km. This pattern occurs in a region of only moderate wind shear, which distinguishes it from the other patterns mentioned before. Obviously, this pattern is related to the 2019–2020 QBO disruption. It seems that the QBO disruption is not so well captured by the reanalyses. This effect is strongest for JRA-55: On the one hand, in late 2019 to mid 2020 the JRA-55 westward wind is too weak at altitudes 20 to 22 km, leading to positive values in Figs. 5c and g, and 6c and g. On the other hand, JRA-55 wind is too westward in the altitude range 22 to 25 km where the QBO disruption is observed, leading to negative values in Figs. 5c and g, and 6c and g. The magnitude of this pattern differs among the stations. For ERA-5 and MERRA-2 similar patterns are being observed, however, much weaker. Still, it should be mentioned that for ERA-5 at Tarawa the pattern is also quite pronounced and almost as strong as for JRA-55.

The fact that this pattern occurs in a region of only moderate wind shear indicates that these differences are not caused by differences in the altitude scales, or different resolutions. Possibly, during the QBO disruption the zonal momentum balance in the reanalyses is not fully realistic, which could be caused by errors in advection, as well as errors in the wave forcing. Obviously, observations that are assimilated in the reanalyses are insufficient to force the models closer to the real state of the atmosphere during the 2019-2020 QBO disruption. Of course, as mentioned above, also in the above mentioned zones of stronger QBO zonal wind shear, deficiencies in the zonal momentum balances will contribute to the observed zonal wind differences between radiosondes and reanalyses.

Next, we focus on the altitude range 10–15 km. In this altitude range we have seen the signature of a strong zonal wavenumber 1 quasi-stationary wave in the radiosonde observations (see Fig. 3). At Belem, in the Western Hemisphere, zonal wind differences between radiosondes and reanalyses are mostly slightly negative, strongest for JRA-55 and MERRA-2, and weakest for ERA-5 (Fig. 5b–d). At the Seychelles and Tarawa, the picture is mixed, while at Mia Padang, in the Eastern Hemisphere, zonal wind differences between radiosondes and the reanalyses are mostly slightly positive (Fig. 6b–d). For Aeolus, the picture is generally mixed. This means that near the equator the quasi-stationary wavenumber 1 is a bit weaker in the reanalyses than in the real atmosphere. These differences, however, will cancel out on zonal average, and, given the relatively large amplitude of the quasi-stationary wavenumber 1 of more than $20\,\mathrm{m\,s^{-1}}$, we consider this only a minor difference.

Still, this raises the question of how well global-scale waves, including also traveling waves, are captured by the reanalyses. Therefore, in the next section we will investigate the zonal wind variances of different global-scale tropical wave modes in

more detail. Because observations of single stations are not suited for this kind of analysis, we will now focus on Aeolus and the reanalyses.

## 4 Variance time series of different global-scale tropical wave modes in Aeolus observations and reanalyses

### 4.1 Symmetric and antisymmetric tropical wave spectra

Equatorially trapped global-scale wave modes can be either symmetric or antisymmetric with respect to the equator (e.g.,
Matsuno, 1966; Wheeler and Kiladis, 1999; Ern et al., 2008). An overview of the theoretical horizontal wave structure for different wave modes is given, for example, in Yang et al. (2003). While, for example, Kelvin waves have their maximum zonal wind amplitude at the equator, several antisymmetric wave modes (for example, mixed Rossby-gravity waves (MRGWs), and antisymmetric equatorial Rossby waves) have their maximum zonal wind amplitudes not at the equator, but at latitudes between the equator and about 15°. This is why we now focus on the wider latitude range 15°S–15°N for performing a windowed 2D
spectral analysis in longitude and time for Aeolus and reanalysis zonal winds, based on a set of overlapping 31-day time windows, similar as in Ern et al. (2008, 2009a) and Ern and Preusse (2009a, b).

For the reanalyses, we perform a 2D fast Fourier transformation (FFT) in longitude and time for a fixed set of latitudes and altitudes with a resolution of 1° latitude × 0.5 km altitude. As we are using 6-hourly model output, the Nyquist frequency for our reanalysis datasets is at $2\,\mathrm{cycles\,day^{-1}}$. Aeolus data are first interpolated on a set of fixed altitudes of 0.25 km vertical
resolution, and then interpolated along the satellite measurement track on a set of fixed latitudes of 1° resolution. Because satellite observations are not on an equispaced rectangular grid in longitude and time a FFT cannot be performed. Instead, for this set of latitudes and altitudes, 2D zonal wavenumber-frequency spectra are calculated by sine-fitting using the same Fourier wavenumbers and frequencies as for the reanalysis data, however, with the limitation that the Aeolus satellite data can unambiguously resolve only frequencies up to about $1\,\mathrm{cycle\,day^{-1}}$. This frequency limit results from the twice-daily asynoptic
satellite sampling by combining ascending and descending parts of the satellite orbit that are measured at different solar local times (e.g., Salby, 1982).

A series of overlapping 31-day time windows is used for calculating the 2D spectra, and for each latitude and altitude the data are detrended separately using a linear fit. The windows are usually slid in time by a step-width of $15$ to $16\,\mathrm{days}$. However, to avoid the data gaps that are present in the Aeolus dataset, the time windows were arranged in a way to avoid these gaps as
much as possible. For a better comparison, it was made sure that the time windows used for Aeolus are a subset of the time windows used for the reanalyses. As the sampling by satellites in low Earth orbit can unambiguously resolve only global-scale waves with zonal wavenumbers up to about 7, and since global-scale waves in the UTLS region are usually dominated by waves of frequencies $< 0.5\,\mathrm{cycles\,day^{-1}}$, we will in the following focus on this reduced spectral domain.

By calculating symmetric and antisymmetric spectra, we are able to separate the symmetric from the antisymmetric wave
modes. As an example, Fig. 7 shows the 2018–2022 average symmetric (Fig. 7a) and antisymmetric (Fig. 7b) spectrum for Aeolus zonal winds at 19 km altitude. The spectra in Fig. 7 represent averages over the latitude range 15°S–15°N. For a more detailed discussion see, for example, Ern et al. (2008). This latitudinal average is performed separately for each of the above

mentioned 31-day time windows, and the spectra shown in Fig. 7 are obtained in a second step by averaging the latitudinally-averaged spectra of all the 31-day time windows. Also shown in Fig. 7 are (assuming zero background wind) the lines of 8,

90, and 2000 m constant equivalent depth from the dispersion relations for several wave modes. A more detailed discussion of these lines is given, for example, in Matsuno (1966), Wheeler and Kiladis (1999), and Ern et al. (2008).

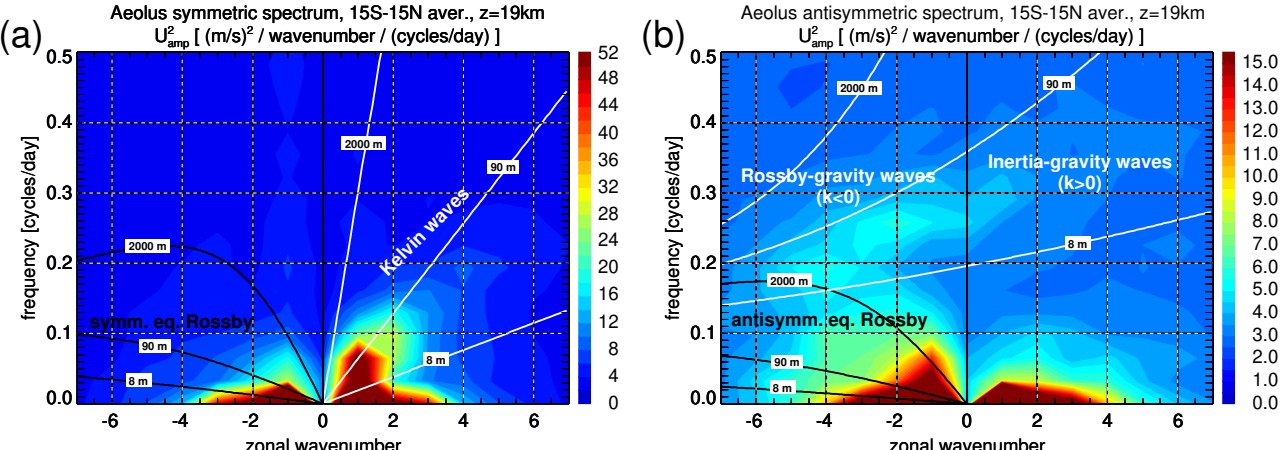

**Figure 7.** Aeolus **(a)** symmetric and **(b)** antisymmetric zonal wind power spectra (i.e., squared spectral amplitudes in units of $m^2 s^{-2}$ wavenumber$^{-1}$ cycles$^{-1}$ day) at 19 km altitude, averaged over the years 2018–2022, and over the latitude range 15°S–15°N. Also shown in (a) are the lines for equivalent depths of 8, 90, and 2000 m from the dispersion relations for Kelvin waves and symmetric equatorial Rossby waves of order n=1. Also shown in (b) are the lines for equivalent depths of 8, 90, and 2000 m from the dispersion relations for MRGWs and n=0 IGWs, as well as for antisymmetric equatorial Rossby waves of order n=2. For the dispersion relation lines zero background wind is assumed. Negative (positive) zonal wavenumbers indicate westward (eastward) propagating waves.

For the symmetric spectrum we show the dispersion lines for symmetric equatorial Rossby waves of order n=1 and for Kelvin waves. For the antisymmetric spectrum the dispersion lines are shown for antisymmetric equatorial Rossby waves of order n=2, for MRGWs, and for a special mode of inertia-gravity waves (of the order n=0) that occupies the continuation of the MRGW

spectral band into positive zonal wavenumbers. (Sometimes this mode is also denoted as eastward MRGWs.) Please note that the spectra will contain also contributions of higher-order equatorial Rossby waves that differ in their meridional structure from the n=1 and n=2 modes, but only the dispersion lines for n=1 and n=2 are shown. As can be seen from Fig. 7, the different wave modes occupy spectral bands that can be relatively well separated. This property will be used in the following for investigating time series of the zonal wind variances, separately for the different wave modes.

In order to preserve the variances of the data contained in the 31-day time windows no tapering has been applied. We did not notice any effects of spectral leakage at higher frequencies. Moreover, we are only interested in relatively low frequencies: <0.15 cycles day$^{-1}$ for equatorial Rossby waves, and <0.3 cycles day$^{-1}$ for Kelvin waves. Only for MRGWs and n=0 IGWs frequencies as high as 0.45 and 0.5 cycles day$^{-1}$, respectively, are used for calculating wave variances in the next subsection. At these low frequencies no indications for spectral leakage are seen in Fig. 7.

## 4.2 Time series of zonal wind variances in different spectral bands

Time series of zonal wind variances are calculated for the different wave modes by integrating over wave bands in the zonal wavenumber-frequency domain that are characteristic for the respective wave mode in the UTLS region. As can be seen from Fig. 7, there is a uniform (white-noise) spectral background that is caused by measurement noise, as well as atmospheric gravity waves that are not resolved by the satellite sampling. This background has to be subtracted from the spectra in order to avoid biases of the variance time series. Otherwise, this background variance would be erroneously attributed to the different global-scale wave modes. For this, we estimate the background as the median over all zonal wavenumbers $k$ with $|k| > 2$, and wave frequencies $\sigma > 0.3\,\mathrm{cycles\,day^{-1}}$. Here, the spectral peaks of diurnal tides at $\sigma = 1\,\mathrm{cycle\,day^{-1}}$ are avoided in the calculation of the spectral background. This estimation is performed separately for the symmetric and the antisymmetric spectra, and also separately for each of the 31-day time windows (i.e., we account for temporal variations of the background). (Please note that in Fig. 7 only part of the unambiguously resolved spectral domain is shown because there are almost no spectral components of significance in the frequency range from 0.5 to somewhat below $1\,\mathrm{cycle\,day^{-1}}$.) For consistency, the same procedure is applied to the reanalyses (although the spectral background in the reanalyses is much lower). For the reanalyses, in addition, the spectral peaks of semidiurnal tides at $\sigma = 2\,\mathrm{cycles\,day^{-1}}$ are avoided for the background estimation.

As shown in Appendix A, zonal wind is a much better parameter for analyzing tropical wave modes in the UTLS than temperature. We also show in Appendix A that, considering the latitude band 15°S–15°N, using the same spectral bands, and assuming the same symmetry as for the zonal winds, meridional wind variances are usually much weaker than zonal wind variances. Therefore, our Aeolus zonal wind variance estimates should be almost unbiased by meridional winds.

### 4.2.1 Symmetric and antisymmetric quasi-stationary wavenumber 1

For the quasi-stationary zonal wavenumber 1, we integrate the power spectra only over the spectral component at $k=1$, and $\sigma = 0\,\mathrm{cycles\,day^{-1}}$. The resulting variances are shown in Fig. 8a, c, e, and g for ERA-5, JRA-55, MERRA-2, and Aeolus, respectively. As can be seen from Fig. 8a, c, e, and g, the symmetric quasi-stationary wave 1 has a huge maximum in the upper troposphere with peak variances of about $150\,\mathrm{m^2\,s^{-2}}$. As the variance of a sine wave averaged over a full period is 0.5 times the squared amplitude of the wave, the peak variances seen in Fig. 8a, c, e, and g correspond to a 15°S–15°N and 31-day average amplitude of square-root two times the variance, i.e., the square-root of $300\,\mathrm{m^2\,s^{-2}}$, which is about $17\,\mathrm{m\,s^{-1}}$. This is roughly in agreement with the findings from the radiosondes in Sect. 3.2, albeit somewhat lower because the selected radiosonde stations are situated close to the equator, i.e., are not representative for the whole latitude band 15°S–15°N. Variances due to higher zonal wavenumber symmetric quasi-stationary waves are usually weaker than about $40\,\mathrm{m^2\,s^{-2}}$, and are therefore not shown here for brevity.

There is a strong seasonality of the symmetric quasi-stationary wavenumber 1 with peak variances during austral and boreal summer, but sometimes there are also enhanced variances in the equinox seasons. These variations are likely caused by a combination of the Walker circulation and the large-scale monsoon circulations. In addition, the El Niño–Southern Oscillation (ENSO) may also play a certain role.

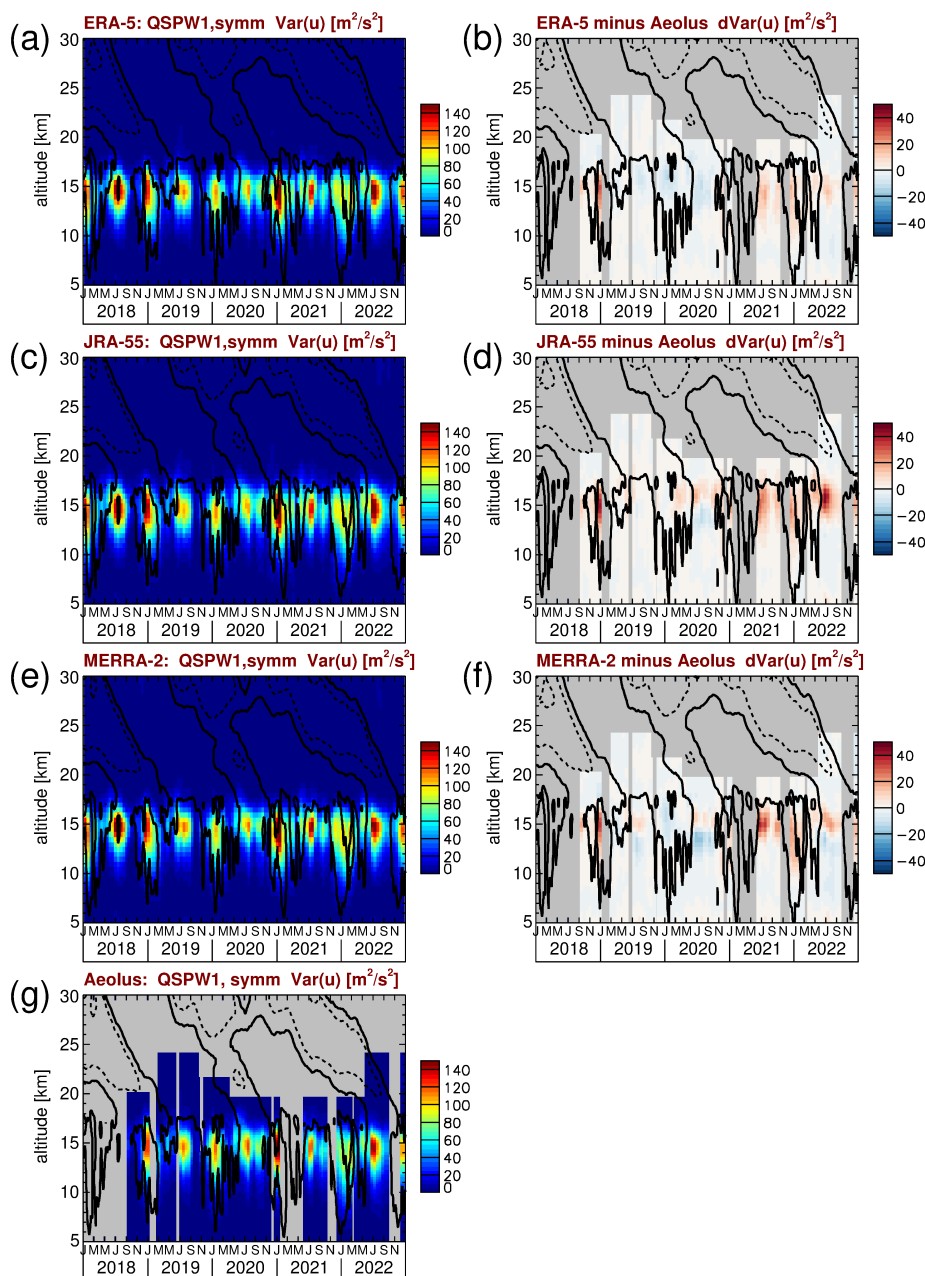

**Figure 8.** Time series of symmetric quasi-stationary zonal wavenumber 1 (QSPW1) zonal wind variances for **(a)** ERA-5, **(c)** JRA-55, **(e)** MERRA-2, and **(g)** Aeolus. Also shown are the differences between **(b)** ERA-5 and Aeolus, **(d)** JRA-55 and Aeolus, and **(f)** MERRA-2 and Aeolus. Contour lines in all panels represent the ERA-5 zonal winds shown in Fig. 1a. Contour lines are plotted every $20\,\mathrm{m\,s^{-1}}$. Dashed (solid) contour lines indicate westward (zero) wind.

The global Walker circulation (named after the findings by Walker and Bliss, 1932) has a strong circulation cell with strong westward winds in the upper troposphere over the Indian Ocean and the Western Pacific, west of about 150°E, while winds in the upper troposphere over the Central and Eastern Pacific are eastward, east of about 150°E until about 100°W (e.g., Eresanya and Guan, 2021). Further circulation cells of the global Walker circulation exist at other longitudes (e.g., Webster and Chang, 1988; Tanaka et al., 2004; Holton and Hakim, 2013). These wind directions in the upper troposphere are qualitatively in agreement with the observed winds at the radiosonde stations in Fig. 3, and contribute to the observed zonal wavenumber 1 structure in the tropical UTLS.

In addition, superimposed on the Walker circulation pattern, also the major large-scale monsoon circulations play an important role for the zonal pattern of zonal winds in the tropical UTLS. These circulations are the dynamical response to the deep convective heating that happens usually during the summer season over the land masses and the Maritime Continent, somewhat displaced from the equator (e.g., Gill, 1980; Holton and Hakim, 2013), which results in large-scale anticyclonal circulations in the UTLS. The large-scale monsoon circulations are important for the upward transport of tracers into the stratosphere (e.g., Park et al., 2009; Konopka et al., 2010; Randel et al., 2010; Ungermann et al., 2016; Vogel et al., 2021, and references therein), and the monsoon regions act as sources for upward-propagating small-scale gravity waves that have strong effect on the dynamics of the middle atmosphere (e.g., Sato et al., 2009; Ern et al., 2013; Thurairajah et al., 2017; Chen et al., 2019; Forbes et al., 2022, and references therein). Particularly the westward directed winds at the southern flank of the Asian summer monsoon contribute to the predominantly westward directed winds in the Eastern Hemisphere (see also, for example, Park et al., 2007, 2009; Konopka et al., 2010), and therefore to the strong zonal wavenumber 1 structure seen in the low-latitude zonal wind in the tropical UTLS. A more detailed investigation, however, including the effect of the other large-scale monsoon systems, is beyond the scope of the current study.

As another effect, ENSO should lead to certain phase shifts of the zonal wind pattern. For example, the predominantly eastward directed winds at Tarawa in the upper troposphere in the years 2020–2022 (see Fig. 3h) could be related to a shift of circulation patterns caused by La Niña conditions in these years (e.g., Yuan and Yan, 2013; WMO, 2022; NOAA, 2023).

Partly, the Madden Julian Oscillation (MJO), having typical wave periods of about 50 days (e.g., Madden and Julian, 1971; Wheeler and Kiladis, 1999), will also project on the quasi-stationary waves because wave periods longer than 31 days are below the frequency resolution given by the length of the time windows used for our Fourier analysis. However, the zonal wind amplitude of the MJO of a few meters per second (e.g., Kiladis et al., 2005) is relatively weak compared to the peak wind variances seen in Fig. 8a, c, e, and g.

In Fig. 8b, d, and f, we calculated the differences between the symmetric quasi-stationary wavenumber 1 variances of the respective reanalysis and Aeolus as a reference. As can be seen from Fig. 8b, d, and f, the differences are usually weaker than about $30 \, \mathrm{m^2 \, s^{-2}}$, i.e., weaker than about 20% of the peak values seen in Fig. 8a, c, e, and g. Overall, this means that there is good agreement between the reanalyses and Aeolus, with ERA-5 performing best. Still, it should be mentioned that in the year 2021 and 2022 the differences between the reanalyses and Aeolus are somewhat stronger, and their characteristics are different as there are stronger peaks of positive deviations with respect to Aeolus that are not seen in the period before, except for late 2018. As mentioned above, in the years 2020–2022, there is a shift of circulation due to the strong La Niña

event that started in 2020. Possibly, the reanalyses are not fully capturing this circulation shift because in the Pacific region, where this circulation shift happens, there are only few observations from radiosondes that can be assimilated in the reanalyses. Consequently, reanalyses might be less reliable in the Pacific region. Still, our data record is far too short to be able to reliably attribute the larger deviations in Fig. 8b, d, and f to ENSO-related phenomena, and our suggested explanation remains very speculative.

Figure 9 shows the same as Fig. 8, but for the antisymmetric quasi-stationary zonal wavenumber 1. Having peak variances of somewhat below $20 \, \mathrm{m^2 \, s^{-2}}$, the antisymmetric quasi-stationary zonal wavenumber 1 is considerably weaker than the symmetric mode. Variances due to higher zonal wavenumber antisymmetric quasi-stationary waves are usually weaker than about $10 \, \mathrm{m^2 \, s^{-2}}$, and therefore not shown for brevity.

It is also remarkable that the seasonality of the quasi-stationary wavenumber 1 antisymmetric mode is different from the seasonality of the symmetric mode. The antisymmetric mode has strong maxima in boreal winter, but only weak maxima in austral winter. Possibly, this seasonality is caused by Rossby waves that are excited in the Winter Hemisphere at midlatitudes and that propagate equatorward. The observed seasonality would agree with the fact that Rossby waves in the Northern Hemisphere during boreal winter are usually much stronger than Rossby waves in the Southern Hemisphere during austral winter. However, a more detailed analysis is beyond the scope of the current study.

Further, it is noteworthy that enhanced variances of the antisymmetric QSPW1 are seen in mid 2019 in the stratosphere in the altitude range 18 to 23 km where significant differences between the zonal winds at the radiosonde stations Belem, Mia Padang, and Tarawa were found in Fig. 4. Enhanced variances are also weakly indicated in Fig 8 for the symmetric QSPW1, albeit at the very lowest levels of the color scale.

We find that deviations between the reanalyses and Aeolus are weaker than about 3 to 4 $\mathrm{m^2 \, s^{-2}}$, i.e., about 20% of the peak values seen in Fig. 9a, c, e, and g. Like before, ERA-5 seems to perform best. Similar as for the symmetric quasi-stationary wavenumber 1, we find the strongest deviations to Aeolus in 2022 and in late 2021. And, as can be seen from Fig. 9b, d, and f, deviations seem to exceed 20% in the mid of the year 2022. Again, these deviations might be related to the ongoing La Niña event and the corresponding shift in circulation. A more detailed analysis, however, is beyond the scope of the current study.

### 4.2.2 Kelvin waves

Figure 10 shows the same as Figs. 8 and 9, but for Kelvin waves. For Kelvin waves, we integrated the symmetric power spectra over the zonal wavenumbers 1 to 6, and over the frequencies $62^{-1} \, \mathrm{cycles \, day^{-1}}$ (i.e., $\approx 0.016 \, \mathrm{cycles \, day^{-1}}$) to $0.3 \, \mathrm{cycles \, day^{-1}}$. Please note that the lower limit of $62^{-1} \, \mathrm{cycles \, day^{-1}}$ is just given for technical reasons and corresponds to half of the frequency resolution of our spectra. We calculate Fourier frequencies in steps of $31^{-1} \, \mathrm{cycles \, day^{-1}}$, which is the frequency resolution given by the length of the 31-day time windows. In order to perform an integration in the spectral domain, an area has to be assigned to the discrete Fourier frequencies, which is given by the spectral resolution in zonal wavenumber and frequency, i.e. "1" for zonal wavenumber and $31^{-1} \, \mathrm{cycles \, day^{-1}}$ for frequency. In particular, this means that for the traveling waves the spectral contributions at zero frequency, i.e., the stationary waves, are not used even though we state for the Kelvin and equatorial Rossby wave modes half the frequency resolution as the lower integration boundary.

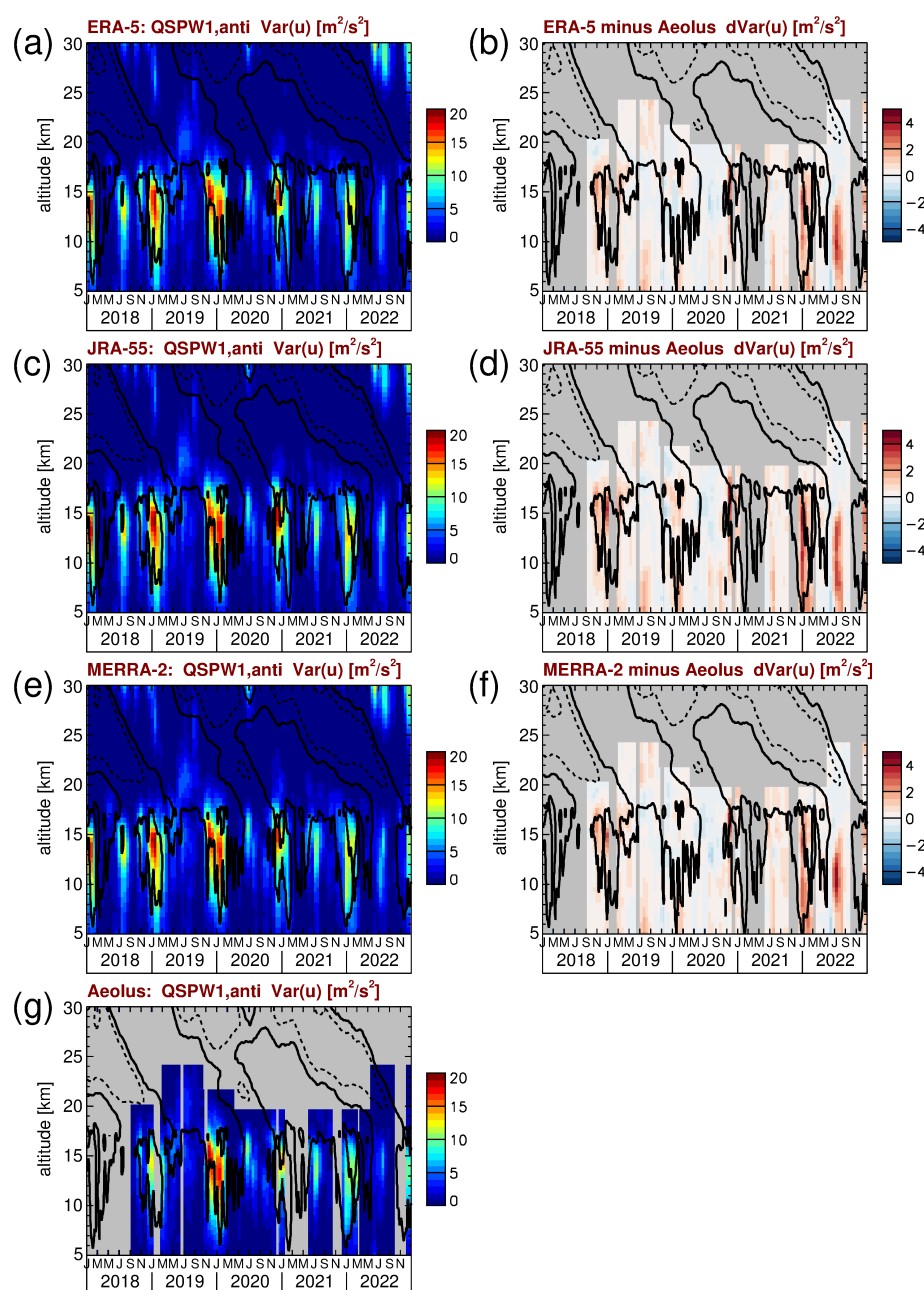

**Figure 9.** Time series of antisymmetric quasi-stationary zonal wavenumber 1 (QSPW1) zonal wind variances for **(a)** ERA-5, **(c)** JRA-55, **(e)** MERRA-2, and **(g)** Aeolus. Also shown are the differences between **(b)** ERA-5 and Aeolus, **(d)** JRA-55 and Aeolus, and **(f)** MERRA-2 and Aeolus. Contour lines in all panels represent the ERA-5 zonal winds shown in Fig. 1a. Contour lines are plotted every 20 m s$^{-1}$. Dashed (solid) contour lines indicate westward (zero) wind.

Similar as for the quasi-stationary waves, Kelvin wave variances are peaking at altitudes of about $15\,\mathrm{km}$, just below the tropical tropopause. Peak values are about $30$–$40\,\mathrm{m^2\,s^{-2}}$, which means that, on $15°$S–$15°$N average, they are the strongest traveling wave mode in UTLS zonal winds. There is also a certain seasonality, but not as strongly linked to the solstice seasons as the quasi-stationary waves.

Kelvin waves contribute significantly to the driving of the QBO eastward phase (e.g., Kawatani et al., 2010; Ern and Preusse, 2009a, b; Kim and Chun, 2015). Therefore, having a good representation of Kelvin waves in models would be highly desirable, in particular in the UTLS region, i.e., close to the sources of the Kelvin waves, in order to have the correct QBO forcing by Kelvin waves already in the lowermost stratosphere.

The differences between the reanalyses and Aeolus (Fig. 10b, d, and f) show that at about $15\,\mathrm{km}$ altitude, where the Kelvin wave variances are highest, the reanalyses have generally lower variances than Aeolus. This indicates that in the upper troposphere Kelvin wave variances are somewhat underrepresented in the reanalyses. This is qualitatively in agreement with the findings by Chien and Kim (2023), who found that precipitation variability due to Kelvin waves is somewhat underestimated in reanalyses. Different from this, at 18 to $19\,\mathrm{km}$, reanalysis Kelvin wave variances tend to be somewhat stronger than those from Aeolus. Generally, however, even though these differences seem to be systematic, deviations are usually weaker than about $4\,\mathrm{m^2\,s^{-2}}$, which means that they are well below 20% of the Kelvin wave peak variances seen in Fig. 10a, c, e, and g. At high altitudes JRA-55 shows the smallest differences to Aeolus. Even though differences between reanalyses and Aeolus are relatively small for Kelvin waves, it was shown by Zagar et al. (2021) that assimilation of Kelvin waves observed by Aeolus significantly improved the forecast errors of the ECMWF model in layers of strong zonal wind shear.

### 4.2.3 Mixed Rossby-gravity waves

Zonal wind variances due to MRGWs are obtained by integrating the power spectra over the zonal wavenumbers $-1$ to $-6$, and over the frequencies between the lines of equivalent depth of $8\,\mathrm{m}$ and $2000\,\mathrm{m}$ for MRGWs shown in Fig. 7b, but limited to frequencies $<0.45\,\mathrm{cycles\,day^{-1}}$. The results are shown in Fig. 11. As can be seen from Fig. 11a, c, e, and g, peak variances of MRGWs are smaller than $4\,\mathrm{m^2\,s^{-2}}$, i.e., much weaker than the wave modes discussed so far. Although zonal wind variances of MRGWs are much weaker than those of Kelvin waves, MRGWs still contribute significantly to the dynamics in the tropics. For example, they play an important role in cyclogenesis (e.g., Dickinson and Molinari, 2002; Frank and Roundy, 2006).

The Aeolus MRGW variances (Fig. 11g) are relatively noisy, because MRGW zonal wind variances are very weak, and Aeolus spectra have a considerable white-noise background. Still, we find reasonable agreement between Aeolus and the reanalyses. Only JRA-55 has lower MRGW variances than ERA-5 and MERRA-2, and also lower variances than Aeolus where Aeolus values are larger and thus more significant. An underrepresentation of MRGWs has been pointed out before for JRA-55 by Harada et al. (2016) for OLR, as well as by Kim et al. (2019) (e.g., their Fig. 6b for meridional wind variances). This can also be seen from the differences between JRA-55 and Aeolus (Fig. 11d). Due to the noise effects, however, the differences in Fig. 11b, d, and f are usually quite large, and sometimes $\sim 1\,\mathrm{m^2\,s^{-2}}$, which is about 50% of the peak values in the altitude range covered by Aeolus.

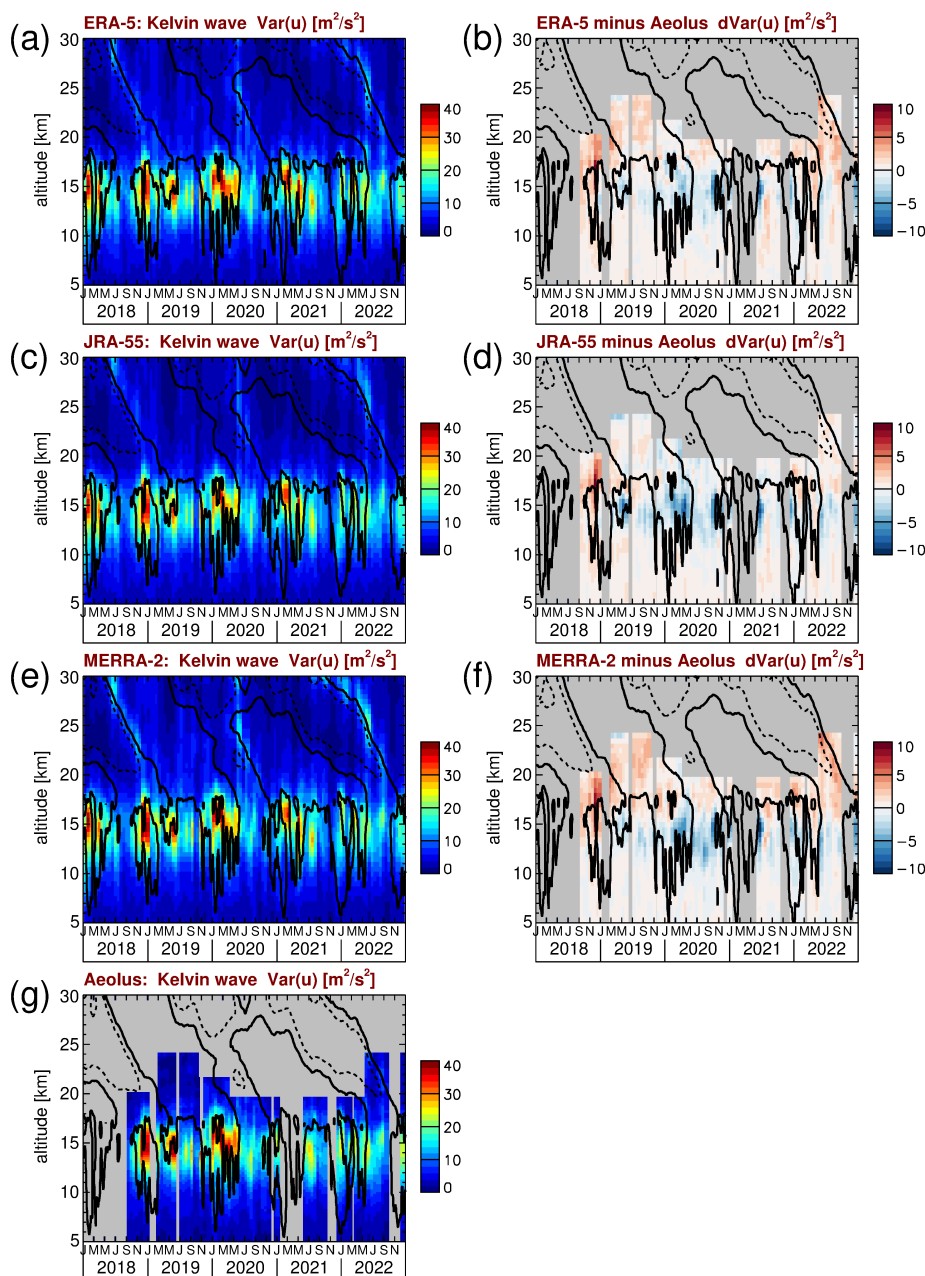

**Figure 10.** Time series of Kelvin wave zonal wind variances for **(a)** ERA-5, **(c)** JRA-55, **(e)** MERRA-2, and **(g)** Aeolus. Also shown are the differences between **(b)** ERA-5 and Aeolus, **(d)** JRA-55 and Aeolus, and **(f)** MERRA-2 and Aeolus. Contour lines in all panels represent the ERA-5 zonal winds shown in Fig. 1a. Contour lines are plotted every $20\,\mathrm{m\,s^{-1}}$. Dashed (solid) contour lines indicate westward (zero) wind.

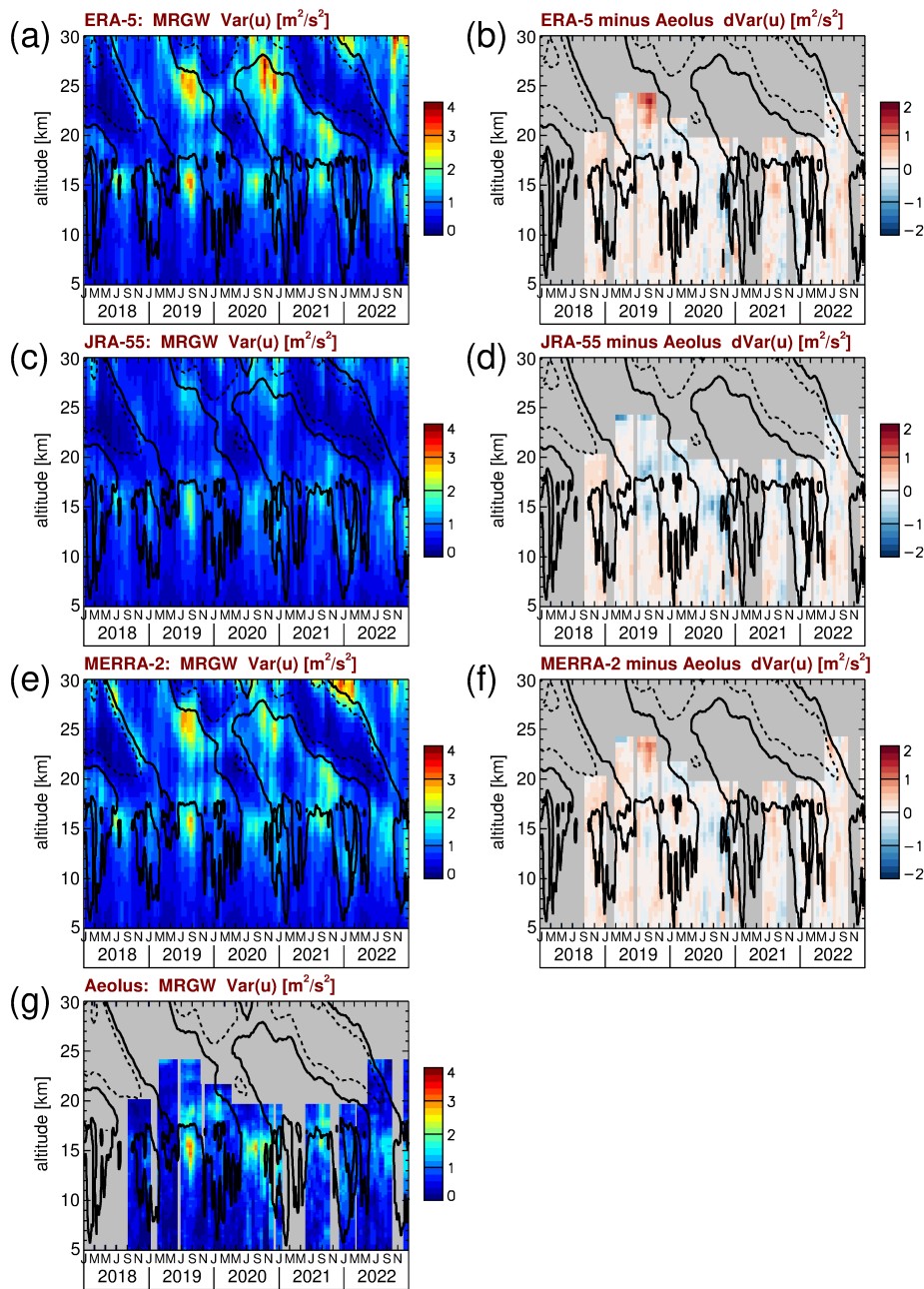

**Figure 11.** Time series of mixed Rossby-gravity wave (MRGW) zonal wind variances for **(a)** ERA-5, **(c)** JRA-55, **(e)** MERRA-2, and **(g)** Aeolus. Also shown are the differences between **(b)** ERA-5 and Aeolus, **(d)** JRA-55 and Aeolus, and **(f)** MERRA-2 and Aeolus. Contour lines in all panels represent the ERA-5 zonal winds shown in Fig. 1a. Contour lines are plotted every $20\,\mathrm{m\,s^{-1}}$. Dashed (solid) contour lines indicate westward (zero) wind.

Further, it should be mentioned that enhanced variances at high altitudes are not necessarily due to equatorial MRGWs, but could also be caused by extratropical Rossby waves propagating equatorward from the winter hemisphere. In particular, enhanced variances around 30 km in January 2022 are suspicious because this variance peak is observed above a QBO westward phase where MRGWs propagating from below could have been filtered out.

### 4.2.4    Symmetric and antisymmetric equatorial Rossby waves

Variances due to equatorial Rossby waves are obtained by integrating over zonal wavenumbers $-1$ to $-6$, and over the frequencies $62^{-1}$ cycles day$^{-1}$ to $0.15$ cycles day$^{-1}$. The results for the symmetric and antisymmetric modes are shown in Figs. 12, and 13, respectively. Peak variances of symmetric and antisymmetric equatorial Rossby waves are similar (about 10–15 m$^2$ s$^{-2}$). The seasonality, however, is different. While both the symmetric and the antisymmetric modes have peak variances around January, the symmetric mode has also significant variances in the whole period shown. Peak values are attained at about 535    14 to 15 km altitude. The increased variances at $\sim$30 km, however, are likely due to extratropical Rossby waves propagating equatorward from the winter hemisphere and are not related to the equatorial Rossby waves.

In the troposphere, equatorial Rossby waves are usually strongly coupled with deep convection and precipitation (e.g., Chen, 2022). Accordingly, convectively coupled equatorial Rossby waves have strong influence on the tropical weather and climate, and, in particular, the monsoons (see also Chatterjee and Goswami, 2004; Janicot et al., 2010). They also contribute to 540    cyclogenesis and the formation of typhoons (e.g., Frank and Roundy, 2006).

As can be seen from the differences between the reanalyses and Aeolus (Fig. 12b, d, and f, and Fig. 13b, d, and f), the reanalyses are performing similarly well. We also find that the reanalysis variances are often somewhat weaker than those from Aeolus at those altitudes and during those periods where variances are strong. This is similar to our finding for Kelvin waves. However, again, deviations are weaker than about 2 m$^2$ s$^{-2}$, and therefore weaker than about 20% of the peak values observed. 545    Therefore, we consider these deviations also to be only minor.

### 4.2.5    n=0 inertia gravity waves

Zonal wind variances due to n=0 IGWs are obtained by integrating over the zonal wavenumbers 1 to 4, and over the frequencies between the lines of equivalent depth of 8 m and 2000 m for n=0 IGWs shown in Fig. 7b, but limited to frequencies $<0.5$ cycles day$^{-1}$.

From Fig. 14a, c, e, and g, we can see that in the altitude range covered by Aeolus peak variances of the IGWs are well below about 1 m$^2$ s$^{-2}$, which is even lower than the variances of MRGWs. Consequently, Aeolus variances are even more noisy than the MRGW variances. Still, we find reasonable agreement between Aeolus and the reanalyses at those altitudes and during those periods where variances are largest. Similar as for the MRGWs, it looks like JRA-55 has considerably weaker variances than ERA-5 and MERRA-2, and also weaker than Aeolus. This can roughly be confirmed from the difference plots shown in 555    Fig. 14b, d, and f. Due to the strong noise of Aeolus variances, however, these difference plots are not very reliable. Differences often are as strong as 0.5 m$^2$ s$^{-2}$, which is close to the peak values found in the altitude range covered by Aeolus.

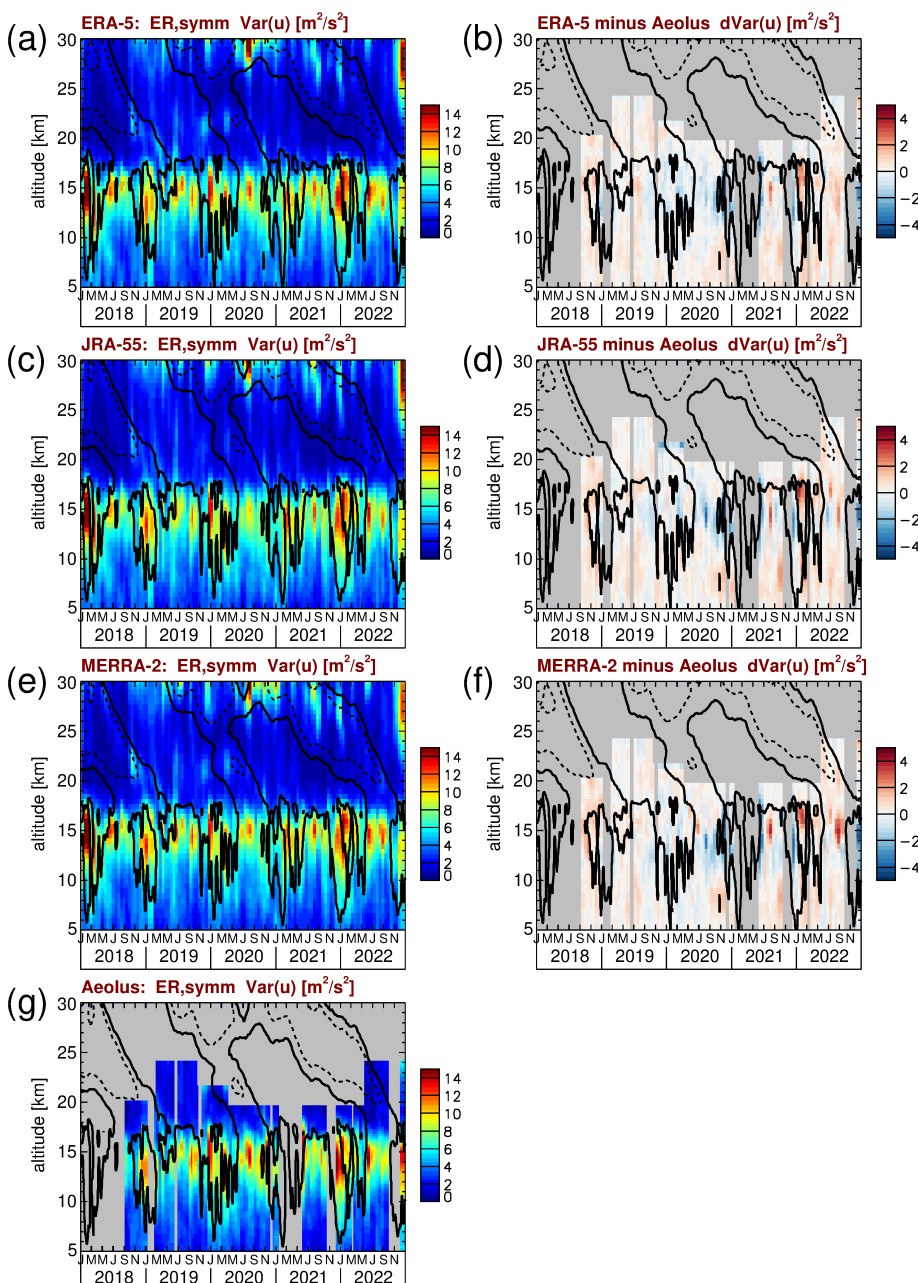

**Figure 12.** Time series of symmetric equatorial Rossby wave (ER) zonal wind variances for **(a)** ERA-5, **(c)** JRA-55, **(e)** MERRA-2, and **(g)** Aeolus. Also shown are the differences between **(b)** ERA-5 and Aeolus, **(d)** JRA-55 and Aeolus, and **(f)** MERRA-2 and Aeolus. Contour lines in all panels represent the ERA-5 zonal winds shown in Fig. 1a. Contour lines are plotted every $20\,\mathrm{m\,s^{-1}}$. Dashed (solid) contour lines indicate westward (zero) wind.

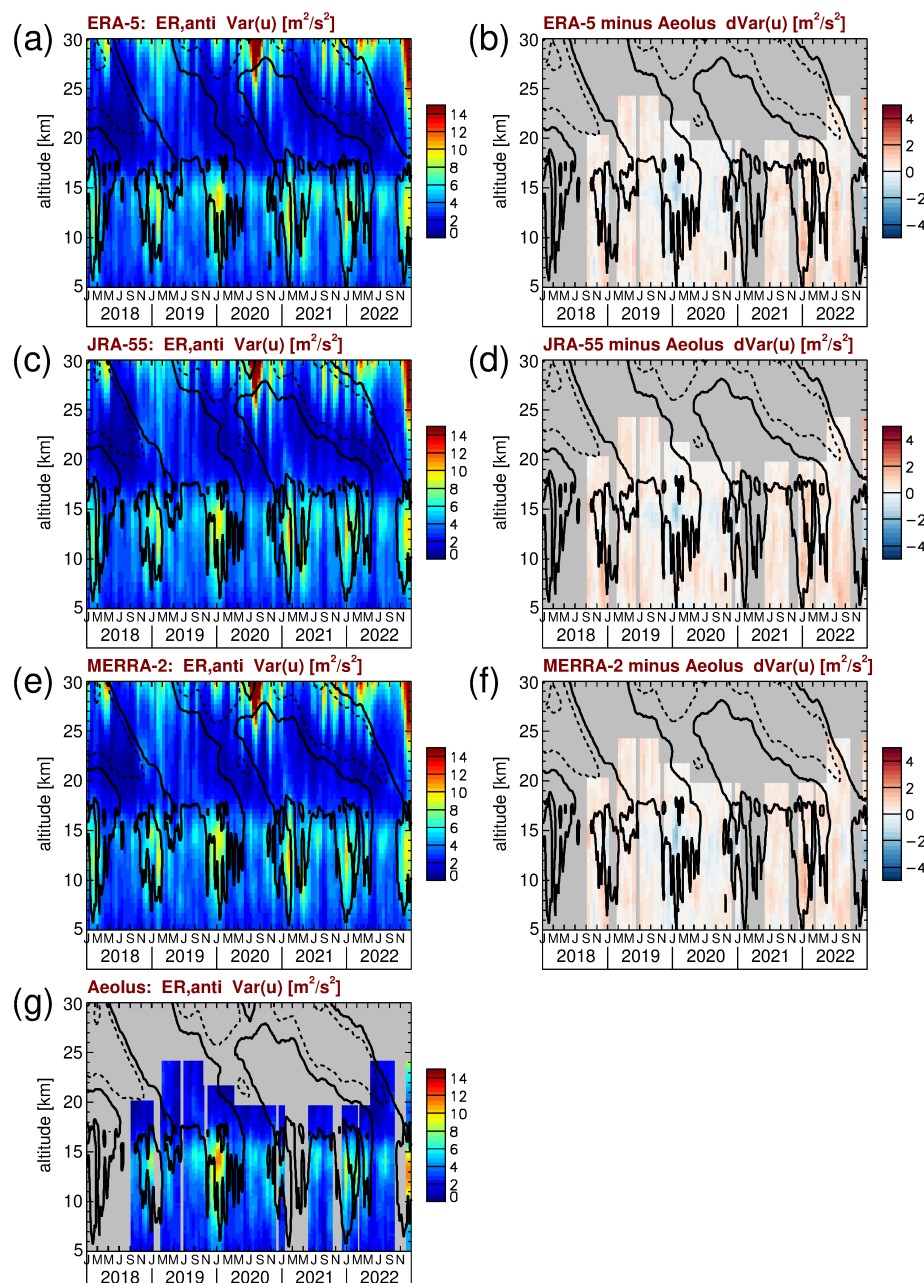

**Figure 13.** Time series of antisymmetric equatorial Rossby wave (ER) zonal wind variances for **(a)** ERA-5, **(c)** JRA-55, **(e)** MERRA-2, and **(g)** Aeolus. Also shown are the differences between **(b)** ERA-5 and Aeolus, **(d)** JRA-55 and Aeolus, and **(f)** MERRA-2 and Aeolus. Contour lines in all panels represent the ERA-5 zonal winds shown in Fig. 1a. Contour lines are plotted every $20\,\mathrm{m\,s^{-1}}$. Dashed (solid) contour lines indicate westward (zero) wind.

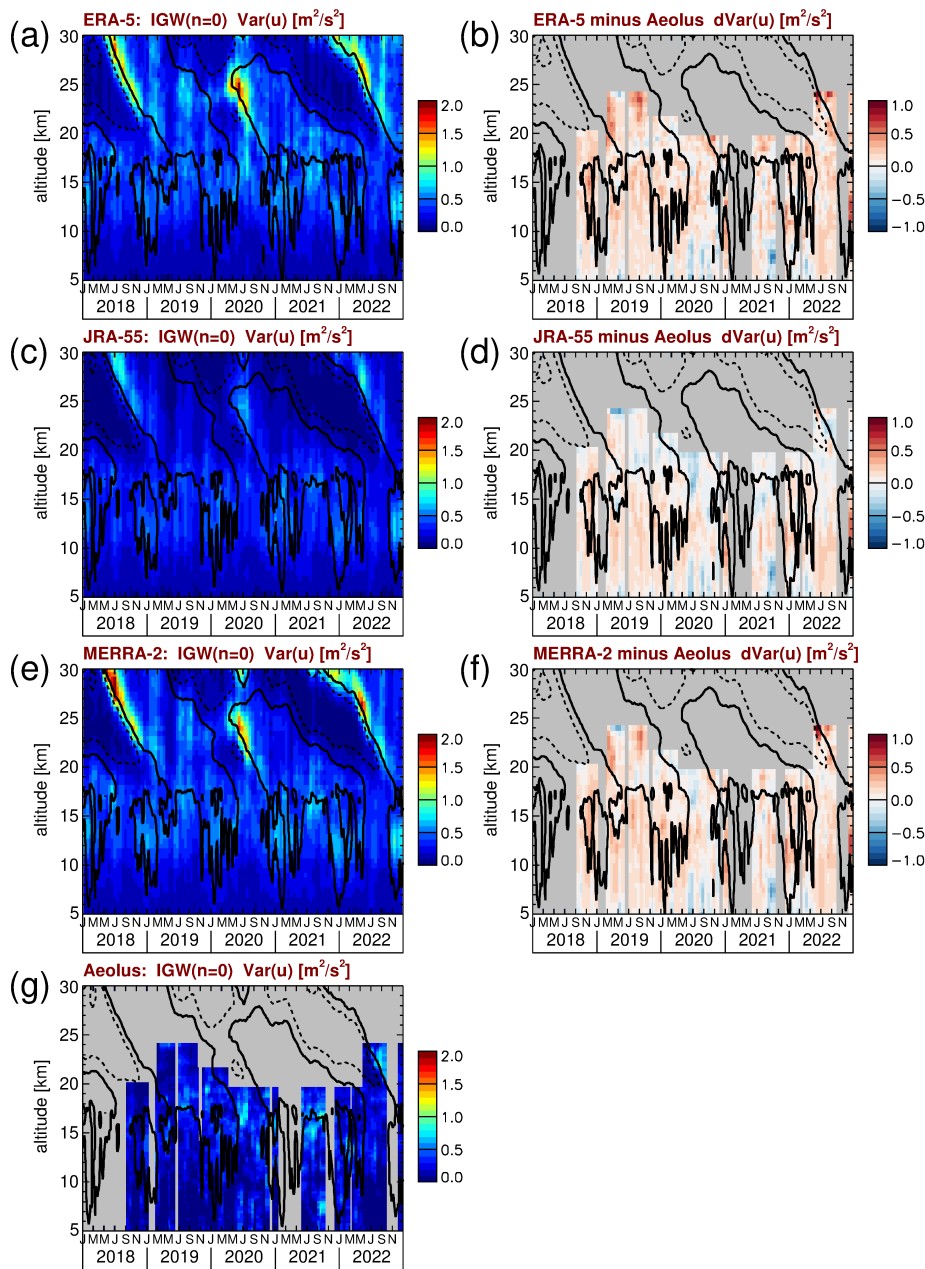

**Figure 14.** Time series of n=0 inertia-gravity wave (IGW (n=0)) zonal wind variances for **(a)** ERA-5, **(c)** JRA-55, **(e)** MERRA-2, and **(g)** Aeolus. Also shown are the differences between **(b)** ERA-5 and Aeolus, **(d)** JRA-55 and Aeolus, and **(f)** MERRA-2 and Aeolus. Contour lines in all panels represent the ERA-5 zonal winds shown in Fig. 1a. Contour lines are plotted every $20\,\mathrm{m\,s^{-1}}$. Dashed (solid) contour lines indicate westward (zero) wind.

## 5 Summary and conclusions

A recent model validation initiative has shown that the QBO and its wave driving in state-of-the-science free-running climate models are not very realistic (Holt et al., 2022; Richter et al., 2022). In particular, it was found by Holt et al. (2022) that climate models have problems to simulate the convectively coupled global-scale tropical waves in the upper troposphere in a realistic way. This is relevant for the driving of the QBO in the lower stratosphere and shows the importance of further validation by global observations of the zonal wind and of the tropical global-scale waves, particularly in the upper troposphere–lower stratosphere (UTLS).

In this study, we have used global wind observations of the Aeolus satellite to investigate the QBO, as well as global-scale tropical wave modes that contribute to the driving of the QBO. For Aeolus, we assume that in the tropics the observed horizontal line of sight winds are representative of the zonal wind, which should be a very good approximation (Krisch et al., 2022). For the global-scale wave modes, the validity of this assumption has been confirmed in Appendix A2. Comparison with three modern reanalyses (ERA-5, JRA-55, and MERRA-2) shows that, on zonal average, the QBO is well represented in all three reanalyses. Averaged over the years 2018–2022, agreement between Aeolus and the reanalyses is better than about $2\,\mathrm{m\,s^{-1}}$, with somewhat larger differences during some periods. The best agreement with Aeolus is found for ERA-5, followed by MERRA-2, and JRA-55.

However, comparison with zonal winds observed at several radiosonde stations in the tropics reveals that, although there is good agreement on zonal average, there can be considerable deviations between reanalyses and radiosonde stations, locally. Some deviations occur in the zonal wind shear zones. These might be caused by slight shifts in the geopotential altitude scales, but increased uncertainties of reanalyses in the shear zones were also found before by, for example, Podglajen et al. (2014) using long-duration balloons in the tropics. Further, deficiencies in the reanalysis momentum budgets may contribute to the deviations in the shear zones. Interestingly, we find also a pattern of deviations in a period of only weak wind shear. These deviations seem to be related to the QBO disruption in 2019–2020, which hints at deficiencies in the momentum budget of the reanalyses also during this period.

The radiosonde stations were selected in a way to cover a longitude range as large as possible. By comparing the zonal wind at the different stations, a strong zonal wavenumber 1 structure in the UTLS could be identified. Further, downward propagating wind anomalies indicate the presence of traveling waves, particularly in the stratosphere. In addition, also in the stratosphere considerable differences in radiosonde winds were found that persisted for several months and may be related to quasi-stationary waves. This shows that adopting winds observed at radiosonde stations as a QBO standard should be treated with caution.

In order to investigate the tropical global-scale waves in more detail, we carried out a windowed Fourier analysis in longitude and time for both the reanalysis and the Aeolus zonal winds. For this, a set of overlapping 31-day time windows was used, and the Fourier analysis was performed for a set of fixed latitudes and altitudes. For a better separation of the different wave modes in the zonal wavenumber-frequency domain, we calculated symmetric and antisymmetric spectra, averaged over the latitude range 15°S–15°N. Because the Aeolus satellite sampling is only able to unambiguously resolve zonal wavenumbers $k$ up to 7,

and since in the UTLS the global-scale wave spectrum is dominated by waves of relatively low frequencies ($\sigma$), we focused on the spectral range $|k| \leq 7$ and $\sigma < 0.5\,\mathrm{cycles\,day^{-1}}$.

Time series of zonal wind variances of the different global-scale equatorial wave modes were obtained by integrating the zonal wavenumber-frequency spectra over the wave bands characteristic for the respective wave mode. As shown in Appendix A, in the UTLS zonal wind is a much better parameter for investigating global-scale equatorial wave modes than temperature, which underlines the importance of global wind observations, such as those from Aeolus. We investigated time series of symmetric and antisymmetric quasi-stationary zonal wavenumber 1, equatorial Kelvin waves, mixed Rossby-gravity waves, symmetric and antisymmetric equatorial Rossby waves, and n=0 inertia-gravity waves.

The strongest wave mode in UTLS zonal wind is the symmetric quasi-stationary zonal wavenumber 1. This wave pattern is likely caused by a superposition of the Walker circulation and the large-scale monsoon circulations, with the El Niño–Southern Oscillation (ENSO) producing certain phase shifts. Among the traveling wave modes, Kelvin waves are the strongest in the UTLS, followed by the equatorial Rossby waves. On 15°S–15°N average, mixed Rossby-gravity waves and n=0 inertia-gravity waves are considerably weaker. It is notable that all wave modes show certain seasonalities. These, however, differ for the different wave modes.

Generally, we find good agreement between all reanalyses and Aeolus zonal wind variances. For most wave modes, the agreement between reanalyses and Aeolus is better than about 20% of the UTLS peak variances. (Please note that 20% of deviation in variances correspond to only 10% of deviation in wave amplitudes, which means an agreement in wave amplitudes of much better than $2\,\mathrm{m\,s^{-1}}$ even for the very strong quasi-stationary wavenumber 1.) Also the seasonality of the different wave modes is well represented in the reanalyses. Only for JRA-55 the mixed Rossby-gravity wave variances, and the n=0 inertia-gravity wave variances seem to be too weak. Further, deviations for the quasi-stationary waves seem to be somewhat larger in the years 2021 and 2022. We speculated that this might be related to the circulation shift by La Niña conditions in these years. However, our data record is far too short to allow more than a vague speculation, and more detailed investigations would be needed.

Although some limitations exist, our findings support the use of the three modern reanalyses considered in our study (ERA-5, JRA-55, and MERRA-2) as a reference for climate models for both the zonal average zonal wind, as well as for the tropical global-scale wave modes. It should, however, be kept in mind that the vertical resolution of Aeolus observations, and of the reanalyses are very similar (about $1$–$2\,\mathrm{km}$). With this altitude resolution, waves of very short vertical wavelength ($< 1\,\mathrm{km}$) that also exist in the tropics (e.g., Bramberger et al., 2022) are therefore not covered, and their contribution to the driving of the QBO still remains an open issue.

*Data availability.* Aeolus wind data were obtained from the ESA Aeolus Online Dissemination System (https://aeolus-ds.eo.esa.int/oads/access/collection/L2B_Wind_Products/tree, ESA (2023c)).

IGRA radiosonde data were obtained from Durre et al. (2023) (https://www.ncei.noaa.gov/data/integrated-global-radiosonde-archive/access/).

The ERA-5 data (https://apps.ecmwf.int/data-catalogues/era5/?class=ea, last access: 03 March 2023, Hersbach et al. (2018)) are available from the European Centre for Medium-Range Weather Forecasts (ECMWF).

MERRA-2 data used in this work are available at the Global Modeling and Assimilation Office (GMAO), MERRA-2 inst3_3d_asm_Nv: 3d, 3-Hourly, Instantaneous, Model-Level, Assimilation, Assimilated Meteorological Fields V5.12.4, Greenbelt, MD, USA, Goddard Earth Sciences Data and Information Services Center (GES DISC), https://doi.org/10.5067/WWQSXQ8IVFW8 (GMAO, Global Modeling and Assimilation Office (2015)).

JRA-55 data used in this work are available at the Japan Meteorological Agency, Japan, 2013, updated monthly. JRA-55: Japanese 55-year Reanalysis, Daily 3-Hourly and 6-Hourly Data. Research Data Archive at the National Center for Atmospheric Research, Computational and Information Systems Laboratory, https://doi.org/10.5065/D6HH6H41 (JMA, Japan Meteorological Agency (2013)).

## Appendix A: Temperature and meridional wind variances for ERA-5

### A1    Temperature variances

For the ERA-5 reanalysis, Fig. A1 shows for several of the wave modes discussed in Sect. 4 temperature variances determined from the same respective wave spectral bands as used before. Figure A1a–c shows variances of quasi-stationary waves and of the equatorial Rossby wave modes, respectively, i.e., variances of Rossby-wave-type modes (see also Matsuno, 1966). Different from this, Fig. A1d–f shows variances of Kelvin waves, MRGWs, and n=0 inertia-gravity waves, respectively, i.e., variances of wave modes that have at least some properties of inertia-gravity waves, (see also Matsuno, 1966). Here we will term these modes "other wave modes" because also the MRGWs are contained which are more of inertia-gravity type for low absolute zonal wavenumbers $|k|$, and more of Rossby-wave-type for high absolute zonal wavenumbers $|k|$. From Fig. A1, it becomes evident that the time series of the variances look quite different from the corresponding time series shown in Figs. 8–14.

For the Rossby-wave-type modes (Fig. A1a–c), different from the zonal wind variances, there are no temperature variance maxima in the altitude range 14 to 15 km. Instead, we often find a double structure of maxima that are located above and below this altitude range. This is likely related to the fact that for Rossby waves the temperature perturbation $T'$ of the wave is proportional to the vertical gradient of the geopotential perturbation $\Phi'$ (see also Sassi et al., 2002; Pancheva et al., 2016; Matthias and Ern, 2018)

$$T' = \frac{H}{R}\frac{d\Phi'}{dz} \tag{A1}$$

with $z$ the log-pressure altitude, $H$ the scale height, and $R$ the ideal gas constant. Assuming geostrophic balance, wind perturbations due to Rossby waves should be related to horizontal gradients of the geopotential perturbation (e.g., Matsuno, 1966; Ern et al., 2013). This means that wind amplitudes and temperature amplitudes of Rossby waves maximize for different conditions. If at a given altitude the geopotential perturbation due to a Rossby wave maximizes, the corresponding strong horizontal geopotential gradients will likely also lead to a maximum of the wind perturbation. Different from this, at the altitude of the maximum geopotential perturbation, the vertical gradient of the geopotential perturbation will be zero. Consequently, the temperature perturbation will be small. This explains why for the Rossby-wave-type modes temperature variances are low where

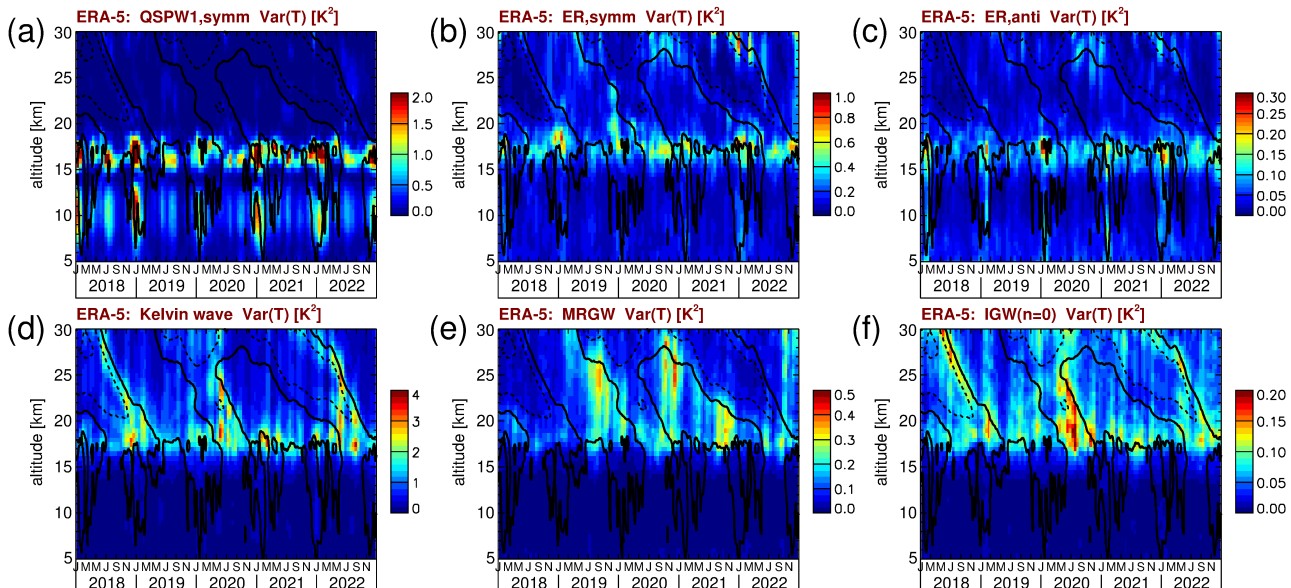

**Figure A1.** ERA-5 temperature variances for **(a)** the symmetric quasi-stationary wave number 1, **(b)** symmetric equatorial Rossby waves, **(c)** antisymmetric equatorial Rossby waves, **(d)** Kelvin waves, **(e)** mixed Rossby-gravity waves, and **(f)** n=0 inertia-gravity waves. The variances are averages over the latitudes $15°$S–$15°$N. Contour lines in all panels represent the ERA-5 zonal winds shown in Fig. 1a. Contour lines are plotted every $20\,\mathrm{m\,s^{-1}}$. Dashed (solid) contour lines indicate westward (zero) wind.

wind variances are maximum. (Of course, this is a simplified picture as equatorial wave modes have also ageostrophic wind contributions (e.g., Matsuno, 1966).)

For the "other wave modes" (Fig. A1d–f), we find that temperature variances are quite low below about $17\,\mathrm{km}$, i.e., below about the altitude of the tropical tropopause. For the Kelvin wave modes, the temperature amplitude and the zonal wind amplitude are approximately related as

$$|\hat{u}| \approx \frac{g}{N} \frac{|\hat{T}|}{T} \tag{A2}$$

where $\hat{u}$ and $\hat{T}$ are the zonal wind and temperature amplitude of the wave, respectively, $g$ is the gravity acceleration, $N$ is the buoyancy frequency, and $T$ is the background temperature (see also Ern et al., 2009b, their Eq. (A13)). At the tropical tropopause the static stability of the atmosphere changes strongly, resulting in a step-like increase of $N$ by a factor for two, or more, from the troposphere to the stratosphere. Accordingly, it is expected that, assuming a constant wind amplitude, the temperature amplitude of a Kelvin wave increases by a factor of about two while propagating from the troposphere into the stratosphere, and, accordingly, temperature variances would increase by a factor of about four. Similar behavior would be expected for the MRGW and the IGW modes because they also have some properties of inertia-gravity waves, and this is indeed seen from Fig. A1d–f.

Of course, the assumption of a constant wind amplitude is somewhat arbitrary and was just made to facilitate the explanation of the temperature amplitude jump at the tropopause. As can be seen from Fig. 10, the zonal wind variances of Kelvin waves reduce by a factor of about 4 between the upper troposphere and the lower stratosphere, which means a reduction of the Kelvin wave zonal wind amplitudes by a factor of about two. Accordingly, a reduction in temperature amplitudes by the same factor would be expected assuming a constant buoyancy frequency (see Eq. (A2). However, as can be seen from Fig. A1d-f, we

observe even the opposite effect: temperature amplitudes increase by a factor of about two between the upper troposphere and the lower stratosphere, which underlines the importance of the effect of changes in the buoyancy frequency at the tropopause.

    Overall, the findings from Fig. A1 illustrate that for investigating variances of equatorial wave modes in the UTLS region zonal wind is much better suited than temperature. This shows the importance of global wind observations, such as Aeolus data.

## A2   Meridional wind variances

Aeolus observes only HLOS winds, and not the full wind vector. Therefore, the question arises whether the zonal wind variances discussed in Sect. 4 might be biased by meridional winds. In order to find out whether our zonal wind variances of global-scale equatorial wave modes observed by Aeolus can be biased by meridional wind components we now investigate the meridional wind variances that are determined by integrating over the meridional wind spectra of ERA-5 using the *same spec-*

*tral bands* as for the zonal winds, and *also the same symmetry* as for the zonal winds (i.e., either symmetric, or antisymmetric with respect to the equator) as performed in Sect. 4.2. In this way, we can find out whether the zonal wind variances of a given wave mode observed by Aeolus can be biased by meridional wind variances of another wave mode.

    Figure A2 shows for several wave modes ERA-5 meridional wind variances determined in the same spectral bands and using the same symmetries that were used for the zonal winds. For a better comparison of magnitudes, the same color scales as in

Sect. 4 are used for the respective wave mode.

    As can be seen from Fig. A2, in the 15°S–15°N averages the meridional wind variances are significantly lower than the zonal wind variances for all wave modes shown, except for the antisymmetric equatorial Rossby waves. For these, meridional wind variances in the same spectral band are sometimes as strong as zonal wind variances. (Please note that these antisymmetric meridional wind variances are attributable to the equatorial Rossby wave mode that is *symmetric* in zonal wind because the

symmetries of zonal and meridional wind are different (e.g. Yang et al., 2003).) However, since in the tropics the sensitivity of Aeolus to zonal wind is about six times higher than for meridional wind, even for this wave mode we do not expect significant biases of the zonal wind variances due to meridional wind components.

    It should also be mentioned that, similarly, the antisymmetric meridional wind variances shown in Fig. A2e for the MRGW spectral band are likely not due to MRGWs, but also due to the equatorial Rossby waves that are symmetric in zonal and

antisymmetric in meridional wind (because MRGWs are symmetric in meridional wind). As has been shown by Kim et al. (2019) the symmetric meridional wind variances due to MRGWs are of the order $2$–$3\,\mathrm{m}^2\,\mathrm{s}^{-2}$ on 15°S–15°N average in the tropopause region at $100\,\mathrm{hPa}$, i.e., considerably stronger than the antisymmetric meridional wind variances shown in Fig. A2e and of about the same magnitude as the antisymmetric zonal wind variances of the MRGWs shown in our Fig. 11.

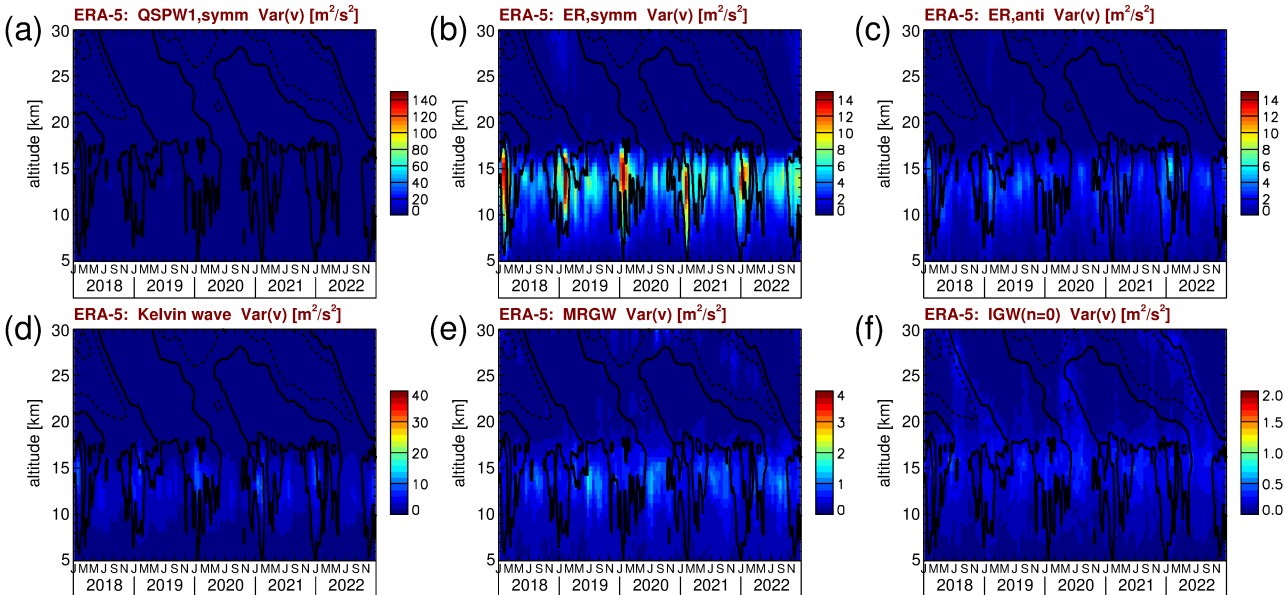

**Figure A2.** ERA-5 meridional wind variances for the spectral wave bands of the **(a)** symmetric quasi-stationary wave number 1, **(b)** symmetric equatorial Rossby waves, **(c)** antisymmetric equatorial Rossby waves, **(d)** Kelvin waves, **(e)** mixed Rossby-gravity waves, and **(f)** n=0 inertia-gravity waves. The variances are averages over the latitudes 15°S–15°N. Contour lines in all panels represent the ERA-5 zonal winds shown in Fig. 1a. Contour lines are plotted every $20\,\mathrm{m\,s^{-1}}$. Dashed (solid) contour lines indicate westward (zero) wind. Please note that the variances shown in this figure are for the same symmetry as for the zonal winds and therefore may be attributable to a different wave mode.

Overall, the fact that the meridional wind variances in Fig. A2 are relatively weak shows that, indeed, the wind variances observed by Aeolus that we attributed to the zonal direction should not be significantly biased by meridional winds.

## Appendix B: Error considerations

### B1   Effects of Aeolus precision

Given a typical Aeolus along-track sampling of ∼90 km, about 5 Aeolus soundings per equator crossing fall into the 2°S–2°N latitude interval used for calculating the winds shown in Fig. 1. According to the number of 15 orbits per day, there are 30 equator crossings per day. Further, our study is based on 7-day average winds. This means that a single Aeolus value shown in Fig. 1 is typically an average over 5×30×7 single soundings, i.e. about 1000 soundings. Assuming a precision of $7\,\mathrm{m\,s^{-1}}$, the precision of the 7-day zonal averages is $7\,\mathrm{m\,s^{-1}}/\sqrt{1000}$, i.e. about $0.2\,\mathrm{m\,s^{-1}}$. This means that in Fig. 1 random errors are almost negligible.

## B2 Effect of vertical wind on Aeolus zonal wind estimates

At altitudes 16–30 km the large scale upwelling in the tropics is weaker than about $0.1\,\mathrm{cm\,s^{-1}}$ (see, for example, Schoeberl et al., 2008). This contribution is much weaker than the zonal winds considered in our study and can therefore be neglected. In the troposphere, large-scale updrafts related to the Walker circulations are typically up to about $0.01\,\mathrm{m\,s^{-1}}$ (e.g., Wang, 2002; Eresanya and Guan, 2021), i.e., also relatively weak. Locally, stronger updrafts and downdrafts can occur particularly in the troposphere during deep convective events. Further strong updrafts and downdrafts can occur in mesoscale and small scale atmospheric gravity waves. The effect of such events should occur as quasi-random fluctuations in the Aeolus HLOS winds and should therefore cancel out in averages. In the zonal wavenumber–frequency spectra, these quasi-random fluctuations should contribute to the uniform spectral background that can be seen, for example, in Fig. 7.

It should however be kept in mind that small-scale fluctuations due to vertical winds should already be strongly suppressed because the Aeolus Rayleigh winds provided as Level 2B data are averages over 86.4 km intervals along-track (e.g., Lux et al., 2020). Therefore, the effect of local thunderstorms should average out. In addition, vertical motions associated with strongly convective cloud systems and large-scale fronts often occur below cloud tops, where Aeolus does not provide observations of Rayleigh winds (see also, for example, Rennie et al., 2022).

## B3 Effect of meridional wind on Aeolus zonal wind estimates

As already stated in Sect. 2.1, in the tropics Aeolus is about 6 times more sensitive to zonal winds than to meridional winds. Therefore biases due to meridional winds should generally be small. In particular, in zonal averages, like those shown in Fig. 1, the meridional wind components of atmospheric waves should cancel out.

Figure B1 shows the meridional wind observed at the eight radiosonde stations introduced in Fig. 2. Like in Fig. 3, the single values are 7-day averages calculated in overlapping 7-day time intervals. As can be seen from Fig. B1, in the stratosphere meridional winds are usually weaker than about $3\,\mathrm{m\,s^{-1}}$. Given the 6 times stronger sensitivity of Aeolus to tropical zonal wind than to tropical meridional wind, biases due to meridional winds should be weaker than about $0.5\,\mathrm{m\,s^{-1}}$. In the troposphere, the 7-day average meridional winds can be stronger (as strong as about 6 to $10\,\mathrm{m\,s^{-1}}$). Therefore, locally biases of the zonal wind of up to 1 to $1.5\,\mathrm{m\,s^{-1}}$ can be expected. On zonal average, however, there should be strong cancellation effects, resulting in much weaker biases.

## B4 Effect of different altitude scales

For the radiosondes and the reanalyses geopotential altitudes are used as altitude coordinates. Of course, there will be differences in the temperature-pressure profiles between the different reanalyses, resulting in slight shifts of the altitude coordinates.

As a measure of uncertainty, for the reanalyses we have calculated at given geopotential altitudes the pressure differences between reanalysis and different radiosonde stations as a reference. These pressure differences were then converted into altitude differences using the local scale height. This has been performed for the dataset of 7-day averages calculated with a time step

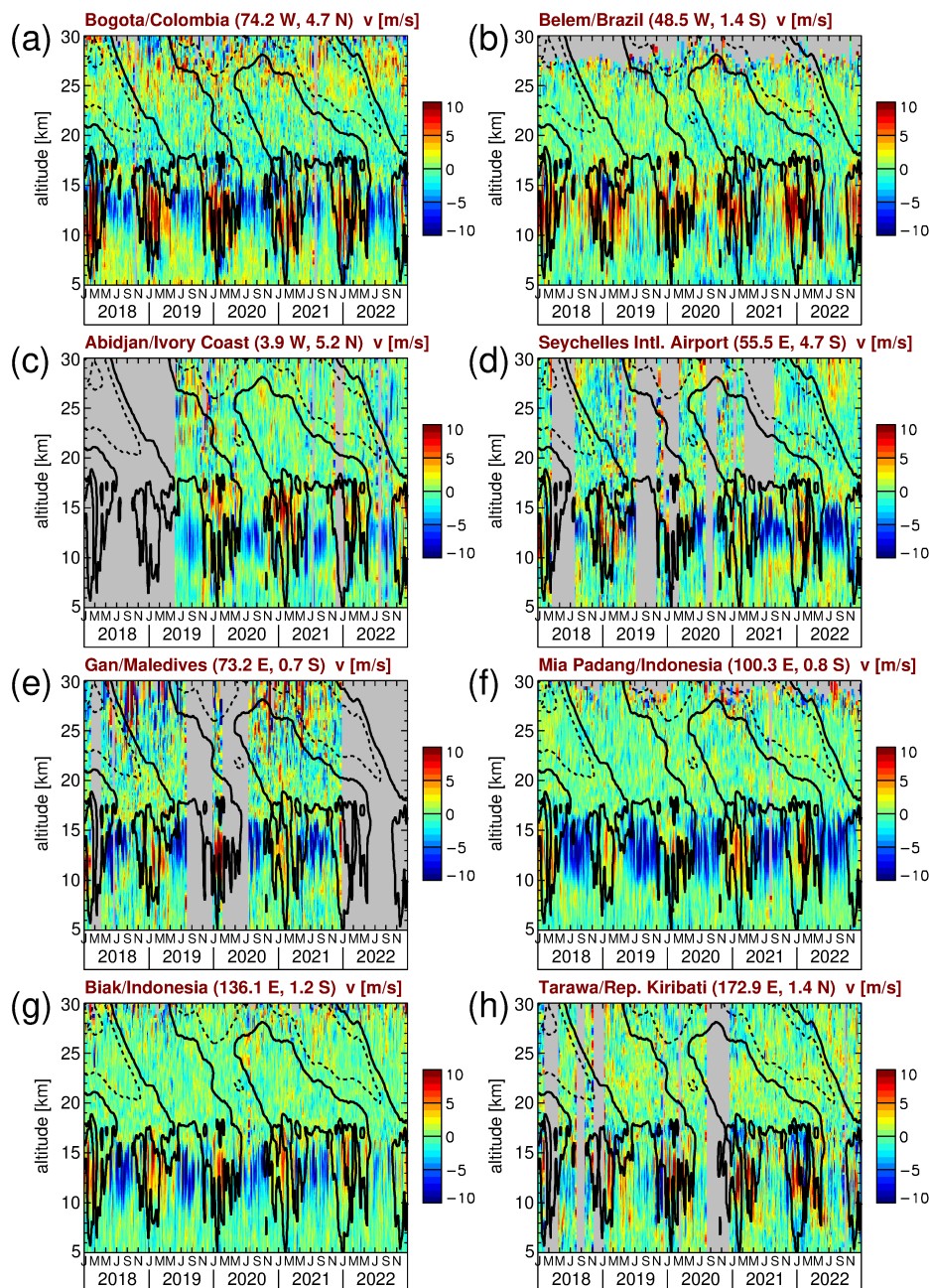

**Figure B1.** Meridional wind observed at the eight radiosonde stations introduced in Fig. 2. Contour lines indicate the ERA-5 zonal mean zonal wind shown in Fig 1a. Contour lines are plotted every $20\,\mathrm{m\,s}^{-1}$. Dashed (solid) contour lines indicate westward (zero) wind.

of three days. The characteristics found for the different radiosonde stations are very similar. Therefore in Fig. B2 we only show the differences for the Tarawa station because the zonal wind differences at Tarawa are relatively strong (see Fig. 6e–h).

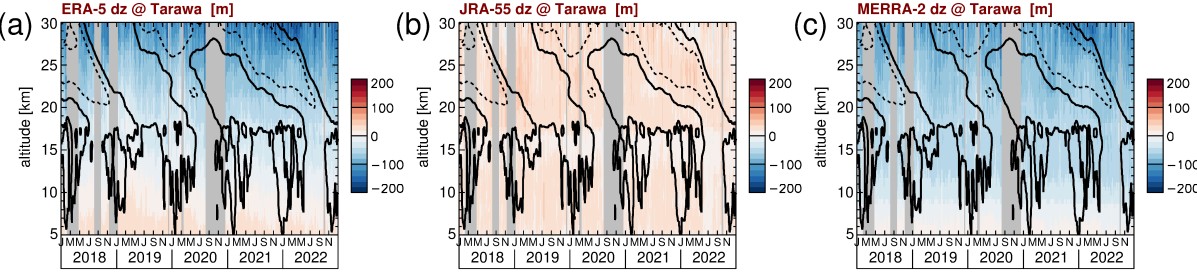

**Figure B2.** Differences in geopotential altitude between **(a)** ERA-5, **(b)** JRA-55, and **(c)** MERRA-2 and the radiosondes at Tarawa as a reference. Altitude differences were calculated from pressure differences at given geopotential altitudes using the local scale height. Contour lines indicate the ERA-5 zonal mean zonal wind shown in Fig 1a. Contour lines are plotted every $20\,\mathrm{m\,s^{-1}}$. Dashed (solid) contour lines indicate westward (zero) wind.

As can be seen from Fig. B2, the altitude differences show a systematic pattern with only small temporal variations. Deviations are typically less than $100\,\mathrm{m}$ with JRA-55 showing the smallest deviations on average (possibly because JRA-55 assimilates IGRA radiosonde data). Given typical magnitudes of QBO-related vertical gradients of the zonal wind of about $10\,\mathrm{m\,s^{-1}\,km^{-1}}$ (perhaps $20\,\mathrm{m\,s^{-1}\,km^{-1}}$ at maximum), this would result in typical biases of 1 to $2\,\mathrm{m\,s^{-1}}$, which is much less than the biases seen for JRA-55 in Figs. 5 and 6. This is some evidence for systematic errors in the zonal momentum budget of JRA-55 being the dominant effect responsible for the wind biases seen in the QBO shear zones. Also for the other reanalyses errors in the zonal momentum budget will contribute to the wind differences seen in the zones of strong wind shear.

It should also be mentioned that Aeolus observations are given on geometric altitudes, and not on geopotential altitudes (see also Rennie et al., 2020). At $\sim20\,\mathrm{km}$ altitude geometric altitudes deviate by about $100\,\mathrm{m}$ from geopotential altitudes. This deviation is of similar magnitude as the differences in geopotential altitude between radiosondes and reanalyses. Therefore, in the QBO shear zones also differences of about 1 to $2\,\mathrm{m\,s^{-1}}$ can be expected when comparing Aeolus with datasets given on geopotential height.

*Author contributions.* ME designed and performed the technical analysis. All coauthors contributed to the interpretation of results and the preparation of the paper.

*Competing interests.* The contact author has declared that none of the authors has any competing interests.

*Acknowledgements.* We would like to thank the teams of the Aeolus instrument, the teams of the various radiosonde stations, as well as the team of the Integrated Global Radiosonde Archive (IGRA) for creating and maintaining the excellent data sets used in our study.

The authors are also grateful to the European Centre for Medium-Range Weather Forecasts (ECMWF), the Global Modeling and Assimilation Office (GMAO) of the National Aeronautics and Space Administration (NASA), and the Japan Meteorological Agency (JMA) for making available the reanalysis data used in our study..

This work was supported by the German Federal Ministry of Education and Research (Bundesministerium für Bildung und Forschung, BMBF) project QUBICC: the work by M. Ern, D. Khordakova, and J. Ungermann was supported by BMBF grant number 01LG1905C, and the work by I. Krisch and O. Reitebuch was supported by BMBF grant number 01LG1905D. QUBICC is part of the Role of the Middle

Atmosphere in Climate II (ROMIC-II) program of BMBF. M. Diallo's research position is funded by the Deutsche Forschungsgemeinschaft (DFG), individual research grant number DI 2618/1-1. This research was supported by the International Space Science Institute (ISSI) in Bern, through ISSI International Team project #567 (Synthetic Gravity Wave Analyses for New Exploitation of Satellite data (SWANS)). Very helpful comments by two anonymous reviewers are gratefully acknowledged.

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
