# Peer review of "The QBO and global-scale tropical waves in Aeolus wind observations, radiosonde data, and reanalyses"

_EGUsphere, 2023_

## Author Comment (AC1)

Many thanks to Referee #1 for appreciating our work and the very helpful comments that will significantly help to improve the manuscript!

Please find below our point-by-point reply to the reviewer concerns. Comments by Reviewer #1 are given in red, our reply is given in black, and changes in the manuscript are indicated in blue.

**Reply to the Main Concerns by Reviewer # 1:**

**(General Comment 1:) Given the following points that are mentioned and discussed in the paper, I am wondering what value of a difference between zonal winds in Aeolus and reanalysis could be considered a (statistically) significant difference:**
**(a) The precision of Aeolus winds is about 3 to 7 m/s (line 104)**
**(b) The vertical and meridional components of winds are ignored (Eq. 3)**
**(c) The winds are interpolated to fixed geopotential heights (Eq. 4).**

The following discussion has been included as a new Appendix B in the revised manuscript:

**(a) Effect of precision:**
Given a typical Aeolus along-track sampling of 90 km, about 5 Aeolus soundings per equator crossing fall into the 2S–2N latitude interval used for calculating the winds shown in Fig. 1 in the manuscript. According to the number of 15 orbits per day, there are 30 equator crossings per day. Further, our study is based on 7-day average winds. This means that a single Aeolus value shown in Fig. 1 is typically an average over 5x30x7 single soundings, i.e. about 1000 soundings. Assuming a precision of $7\,\mathrm{m\,s^{-1}}$, the precision of the 7-day zonal averages is $7\,\mathrm{m\,s^{-1}}/\sqrt{1000}$, i.e. about $0.2\,\mathrm{m\,s^{-1}}$. This means that in Fig. 1 random errors are almost negligible.

**(b1) Effect of vertical wind:**
At altitudes 16–30 km the large scale upwelling in the tropics is weaker than about $0.1\,\mathrm{cm\,s^{-1}}$ (see Schoeberl et al., 2008). This contribution is much weaker than the zonal winds considered in our study and can therefore be neglected. In the troposphere, large-scale updrafts related to the Walker circulations are typically up to about $0.01\,\mathrm{m\,s^{-1}}$ (e.g., Wang, 2002; Eresanya and Guan, 2021), i.e., also relatively weak. Locally, stronger updrafts and downdrafts can occur particularly in the troposphere during deep convective events, or generally in atmospheric gravity waves. The effect of such events should occur as quasi-random fluctuations in the Aeolus HLOS winds and should therefore cancel out in averages. In the zonal wavenumber–frequency spectra, they should contribute to the uniform spectral background that can be seen, for example, in former Fig. 6.
It should however be kept in mind that small-scale fluctuations due to vertical winds should already be strongly suppressed because the Aeolus Rayleigh winds provided as Level 2B data are averages over 86.4 km intervals along-track (Lux et al., 2020). Therefore the effect of local thunderstorms should average out. In addition, vertical motions associated with strongly convective cloud systems and large-scale fronts often occur below cloud tops, where Aeolus does not provide observations of Rayleigh winds (see also Rennie et al., 2022).

**(b2) Effect of meridional wind:**
As already stated in Sect. 2.1 in the manuscript, in the tropics Aeolus is about 6 times more sensitive to zonal winds than to meridional winds. Therefore biases due to meridional winds should generally be small. In particular, in zonal averages, like those shown in Fig. 1 in the manuscript, the meridional wind components of atmospheric waves should cancel out.
In the new Appendix B we have included time series of 7-day average meridional winds observed at the 8 radiosonde stations considered in our study. In the stratosphere, these meridional winds are usually weaker than about $3\,\mathrm{m\,s^{-1}}$. Given the 6 times stronger sensitivity of

Aeolus to tropical zonal wind than to tropical meridional wind, biases due to meridional winds should be weaker than about $0.5\,\mathrm{m\,s^{-1}}$. In the troposphere, the 7-day average meridional winds can be stronger (as strong as about 6 to $10\,\mathrm{m\,s^{-1}}$). Therefore locally biases of the zonal wind of up to 1 to $1.5\,\mathrm{m\,s^{-1}}$ can be expected. On zonal average, however, there should be strong cancellation effects, resulting in much weaker biases.

[Figure]

**Figure 1:** Meridional wind observed at the eight radiosonde stations introduced in Fig. 2 in the manuscript. Contour lines indicate the ERA-5 zonal mean zonal wind shown in Fig. 1a in the manuscript.

**(c) Interpolation to fixed geopotential heights:**

Of course, there will be differences in the temperature-pressure profiles between the different reanalyses, resulting in slight shifts of the altitude coordinates.

As a measure of uncertainty, for the reanalyses we have calculated at given geopotential altitudes the pressure differences between reanalysis and different radiosonde stations as a reference. These pressure differences were then converted into altitude differences using the local scale height. This has been performed for the dataset of 7-day averages calculated with a time step of three days. The characteristics found for the different radiosonde stations are very similar. Therefore we only show the differences for the Tarawa station because the zonal wind differences at Tarawa are relatively strong (see former Fig. 5e–h).

[Figure]

**Figure 2:** Differences in geopotential altitude between **(a)** ERA-5, **(b)** JRA-55, and **(c)** MERRA-2 and the radiosondes at Tarawa as a reference. Altitude differences were calculated from pressure differences at given geopotential altitudes using the local scale height. Zonal wind contour lines are from ERA-5. Contour interval is $20\,\mathrm{m\,s^{-1}}$, westward winds are indicated by dashed contour lines, and zero wind is indicated by solid contour lines.

As can be seen from this figure, the altitude differences show a systematic pattern with only small temporal variations. Deviations are typically less than $100\,\mathrm{m}$ with JRA-55 showing the smallest deviations on average (possibly because JRA-55 assimilates IGRA radiosonde data). Given typical magnitudes of QBO-related vertical gradients of the zonal wind of about $10\,\mathrm{m\,s^{-1}\,km^{-1}}$ (perhaps $20\,\mathrm{m\,s^{-1}\,km^{-1}}$ at maximum), this would result in typical biases of 1 to $2\,\mathrm{m\,s^{-1}}$, which is much less than the biases seen for JRA-55. This is some evidence for systematic errors in the zonal momentum budget of JRA-55 being the dominant effect responsible for the wind biases seen in the QBO shear zones. Also for the other reanalyses errors in the zonal momentum budget will contribute to the wind differences seen in the zones of strong wind shear.

It should also be mentioned that Aeolus observations are given on geometric altitudes (see also Rennie et al., 2020). At $\sim$20 km altitude geometric altitudes deviate by about $100\,\mathrm{m}$ from geopotential altitudes. This deviation is of similar magnitude as the differences in geopotential altitude between radiosondes and reanalyses. Therefore, in the QBO shear zones also differences of about 1 to $2\,\mathrm{m\,s^{-1}}$ can be expected when comparing Aeolus with datasets given on geopotential height.

**(General Comment 2) Has any kind of tapering been applied to the 31-day period windows before applying the 2D Fourier transforms? If not, do you see any spectral leakage at higher frequencies?**

In order to preserve the variances of the data contained in the 31-day period windows no tapering has been applied. We did not notice any effects of spectral leakage at higher frequencies. It can also be seen from the spectra in former Fig. 6 that there are no indications for spectral leakage. In addition, it should be kept in mind that in this study we are only interested in relatively low frequencies: $<0.15\,\mathrm{cycles\,day^{-1}}$ for equatorial Rossby waves, and $<0.3\,\mathrm{cycles\,day^{-1}}$ for Kelvin waves. Only for MRGWs and n=0 IGWs frequencies as high as 0.45 and $0.5\,\mathrm{cycles\,day^{-1}}$, respectively, are used.
This information has been added in the revised manuscript at the end of Sect. 4.1.

**(General Comment 3)** Related to the point above, when a 31-day window is being used, and assuming no tapering in time, the lowest resolved frequency should be 0.03 cycles/day. However, in isolating the Kelvin waves, the frequency of 0.015 cycles/day has been used as the lower band (i.e., a period of 67 days). I think this results in some contamination of the Kelvin wave signal by the quasi-stationary waves.

We calculate Fourier frequencies in steps of $31^{-1}$ cycles day$^{-1}$, which is the frequency resolution given by the length of the 31-day time windows. The number of $0.015$ cycles day$^{-1}$ is just given for technical reasons and corresponds to half of the frequency resolution of our spectra. In order to perform an integration in the spectral domain, an area has to be assigned to the discrete Fourier frequencies, which is given by the spectral resolution in zonal wavenumber and frequency, i.e. "1" for zonal wavenumber and $31^{-1}$ cycles day$^{-1}$ for frequency. In particular, this means that for the traveling waves the spectral contributions at zero frequency, i.e., the stationary waves, are not used even though we state for the Kelvin and equatorial Rossby wave modes half the frequency resolution as the lower integration boundary.

This information has been added in the revised manuscript in Sect. 4.2.2. Further, we now state the more correct value of $62^{-1}$ cycles day$^{-1}$ that was actually used (half the frequency resolution) as the lower frequency boundary for the integration domains of the Kelvin and equatorial Rossby waves.

**Reply to the Specific Comments by Reviewer # 1:**

**(1) Line 52: It might be useful to mention that the gravity waves are usually not resolved in climate models and their effects are approximated using different parameterization schemes.**

As recommended by the reviewer, this is now stated in the revised manuscript.

**(2) Lines 130 to 150: My understating is that the Aeolus and radiosondes winds are interpolated to 0.25 km vertical resolution, but the reanalysis are interpolated to a 0.5 vertical resolution. If this is the case, then how you subtract them in the next sessions?**

The differences are calculated only for the altitude levels that are common to both data sets.

This is now stated in the revised manuscript in the first paragraph of Sect. 2.3.

**(3) Figure 1: I am not sure why the one standard deviation is shown with respect to zero. I think it should be shown with respect to the mean differences (i.e., the red line).**

The intention of Fig. 1 in the manuscript is to show both the systematic deviations (on long-term average) and the standard deviation with Aeolus as a reference. For this purpose, the standard deviation is calculated with respect to the zero line, i.e. Aeolus as a reference. It was not our intention to show how robust (in a statistical sense) the systematic differences between Aeolus and the reanalyses are.

This is now stated more clearly in the revised manuscript.

**(4) Figure 6: Have you tried using a log-scale for plotting the power? I think it would be useful in extracting the interesting features in higher frequencies.**

Linear scales were used intentionally to show the features of the MRGWs and n=0 IGWs at higher frequencies. As can be seen from former Fig. 6, the problem is that in the antisymmetric spectrum there is a relatively high uniform spectral background of about $3.5\,\mathrm{m^2\,s^{-2}}$/wavenumber/(cycles/day). The peak spectral densities of MRGWs and IGWs are at about 5 and 4.5, respectively, in the same units, i.e., not much higher. Using a logarithmic scale would therefore obscure the MRGW and IGW signals.

No changes were made in the revised manuscript.

**(5) Section 4.2.4. You might want to mention that the peak variances at 30 km are due to extratropical Rossby waves propagating equatorward from the winter hemisphere and are not related to the equatorial Rossby waves.**

This is now mentioned in the revised manuscript.

**(6) Figure A2: I am a little bit surprised by the panel for MRGW. I expected to see a stronger signal in meridional wind than in the zonal wind for MRG waves. In isolating the MRG signal based on the wavenumber-frequency spectrum, have you considered the fact the MRG waves are symmetric in meridional wind (while they are antisymmetric in zonal wind)?**

We are sorry for not stating explicitly enough the intention of Appendix A2!
Figure A2 was meant to show that meridional wind components in the same spectral band and of the same symmetry as the zonal wind variances shown in Sect. 4.2 will not significantly contaminate the zonal wind variances shown in Sect. 4.2. This means that, for this purpose,

we have to integrate over the same spectral bands as for the zonal wind variances, and **we also have to use the same symmetry as for the zonal winds**. This is why we have to investigate the antisymmetric contribution of the meridional wind variances in the MRGW spectral wave band. Of course, this contribution is relatively small because MRGWs are symmetric in meridional winds. The same holds for the n=0 IGWs. Further, it should be mentioned that the meridional wind variances in Fig. A2b are so strong because the symmetric meridional wind signal of the equatorial Rossby wave mode that is antisymmetric in zonal wind, but symmetric in meridional wind is picked up. Still, this meridional wind signal would not significantly affect the zonal wind variances of the symmetric (in zonal wind) equatorial Rossby waves. Similarly, the meridional wind variances in Fig A2e (MRGWs) are likely due to the antisymmetric meridional wind signal of the equatorial Rossby wave mode that is symmetric in zonal wind.

Indeed, for MRGWs symmetric meridional wind variances are somewhat stronger than the antisymmetric zonal wind variances. For example, Kim et al. (2019) find meridional wind variances between 2 and $3\,\mathrm{m}^2\,\mathrm{s}^{-2}$ on 15S–15N average for different reanalyses in the tropopause region at 100 hPa. These numbers are similar to the zonal wind peak variances found in our study. Further, also Kim et al. (2019) find that, compared to other reanalyses, JRA-55 has relatively weak MRGW variances.

This more detailed discussion has been included in the revised Appendix A2.

**(7) Line 207: The comma before "Also", should change to a dot.**

done

**References**

Eresanya, E. O. and Guan, Y.: Structure of the Pacific Walker Circulation depicted by the reanalysis and CMIP6, Atmosphere, 12, 1219, https://doi.org/10.3390/atmos12091219, 2021.

Kim, Y.-H., Kiladis, G. N., Albers, J. R., Dias, J., Fujiwara, M., Anstey, J. A., Song, I.-S., Wright, C. J., Kawatani, Y., Lott, F., and Yoo, C.: Comparison of equatorial wave activity in the tropical tropopause layer and stratosphere represented in reanalyses, Atmos. Chem. Phys., 19, 10027–10050, https://doi.org/10.5194/acp-19-10027-2019, 2019.

Lux, O., Lemmerz, C., Weiler, F., Marksteiner, U., Witschas, B., Rahm, S., Geiß, A., and Reitebuch, O.: Intercomparison of wind observations from the European Space Agency's Aeolus satellite mission and the ALADIN Airborne Demonstrator, Atmos. Meas. Tech., 13, 2075–2097, https://doi.org/10.5194/amt-13-2075-2020, 2020.

Rennie, M., Tan, D., Andersson, E., Poli, P., Dabas, A., de Kloe, J., Marseille, G.–J., and Stoffelen, A.: Aeolus Level-2B algorithm theoretical basis document (Mathematical description of the Aeolus L2B processor), ECMWF report, AED-SD-ECMWF-L2B-038, Version: 3.40, available at: https://earth.esa.int/eogateway/documents/20142/37627/Aeolus-L2B-Algorithm-ATBD.pdf (last access: 06 June 2023), 2020.

Rennie, M., Healy, S., Abdalla, S., McLean, W., and Henry, K.: Aeolus positive impact on forecasts with the second reprocessed dataset, ECMWF newsletter, No. 173, 14–20, doi:10.21957/mr5bjs29fa, 2022.

Schoeberl, M. R., Douglass, A. R., Stolarski, R. S., Pawson, S., Strahan, S. E., and Read, W.: Comparison of lower stratospheric tropical mean vertical velocities, J. Geophys. Res., 113, D24109, doi:10.1029/2008JD010221, 2008.

Wang, C.: Atmospheric circulation cells associated with the El Nino–Southern Oscillation, J. Climate, 15, 399–419, 2002.

---

## Author Comment (AC2)

Many thanks to Referee #2 for carefully reading the paper and for the very constructive comments that will significantly help to improve the manuscript!

Please find below our point-by-point reply to the reviewer concerns. Comments by Reviewer #2 are given in red, our reply is given in black, and changes in the manuscript are indicated in blue.

**Reply to the Specific Comments by Reviewer # 2:**

**(1) L36 : "one of the main ..." : What would be other processes in the tropics ?**

Other source processes could be shallow convection, or, in the case of small scale gravity waves, for example, wind shear and excitation by orography. The present study, however, is not focused on gravity waves, and even for gravity waves in the tropics source processes other than convection should be less prominent.
Therefore, in order to avoid mentioning too many details and to stay more focused, the sentence was reworded as follows:

"...with convection as the main wave generation process."

**(2) L43 : "... MRGWs" : It may be arguable whether the MRGW is a prominent wave mode in zonal winds, as also seen in the result of this paper.**

Traditionally, MRGWs were believed to be important. However, as indicated in this study, MRGWs seem to be much weaker than Kelvin waves. Therefore we revised the text as follows:

"... One of the most prominent wave modes in zonal winds are equatorial Kelvin waves. In zonal winds, Kelvin waves are symmetric with respect to the equator. Another wave mode are mixed Rossby-gravity waves (MRGWs). The latter are antisymmetric in zonal winds. Further, there are..."

**(3) L48 : "in the troposphere" : Isn't the OLR a quantity at the top of atmosphere, observed from the space ?**

Indeed, this formulation may be somewhat misleading. Here "in the troposphere" was addressing the wave distributions in the troposphere — not where the observations were made. Therefore the text was reworded as follows:

"Global observations from satellite are particularly suited for these kind of investigations: An overview of the spectral contributions of the different wave modes in the troposphere is given, for example, by Wheeler and Kiladis (1999) based on outgoing longwave radiation (OLR) observations. In the stratosphere, the spectral contributions of the different wave modes were investigated, for example, by Ern et al. (2008) using global temperature observations from space."

**(4) L51 : I would suggest including Bushell et al. (2022, doi:10.1002/qj.3765) too.**

reference has been added

**(5) L61 : Could you please add a reference ?**

Three references have been added: Mote et al. (1996), Pumphrey et al. (2008), and Butchart (2014)

**(6) L52 : Holt et al. should be removed in this line (but kept in L53) as they did not analyze gravity waves but resolved waves (mainly Kelvin waves and MRGWs).**

done

**(7) L82 : "models" − > "reanalyses"**

changed as suggested

**(8) L154, 159 : "T639", "T319" − > "TL639", "TL319"**

changed as requested

**(9) L162 : "∼21 km in the tropics"**

changed as suggested

**(10) Figures : Reference mean-wind contours : Why has MERRA-2 been chosen (while the difference of the reanalysis outputs from Aeolus winds is minimum for ERA-5) ?**

At the time of writing only JRA-55 and MERRA-2 were available until end 2022.

The ERA-5 dataset has now been updated until end of 2022. As recommended, we now use ERA-5 contour lines as a reference.

**(11) L236–237 : Would traveling waves remain in monthly mean (which is usually used for the QBO) ?**

This question addresses the point whether local radiosonde stations would be a good proxy for the QBO. Indeed, the effect of traveling waves would be much reduced in monthly averages. Still, it is unclear whether local stations can serve as a QBO proxy, and obviously this topic needs some more discussion.

Therefore we have included another figure in the manuscript that shows the zonal wind differences between all other radiosonde stations and Mia Padang as a reference. Mia Padang is selected as a reference because it is relatively close to the equator and has a good temporal coverage. We have also added the following detailed discussion in the manuscript.

Of course, there are large zonal wind differences in the troposphere that are related to the quasi-stationary wave pattern.

The new figure also shows that in the stratosphere for the three stations that are located about 5° off-equator (Bogota, Abidjan, and the Seychelles) there are persistent zonal wind differences of up to about $10\,\mathrm{m\,s^{-1}}$ over several months. This means that the zonal wind observed at these stations would not be a good QBO proxy, and stations closer to the equator would be preferable.

However, in the stratosphere there are also considerable zonal wind differences between Mia Padang and the other stations that are located close to the equator. Partly, these differences are caused by traveling global-scale waves. This effect is most strongly seen in the differences between Belem and Mia Padang — probably because the difference in longitude between these two stations is relatively large such that zonal wavenumber 1 waves are almost in antiphase at these two stations. As most of the traveling waves seem to have periods of 30 days, or shorter, the effect of these waves will be strongly reduced in monthly averages.

Interestingly, there are also differences between Belem and Mia Padang that persist for several months. For example, in 2019 between 18 and 25 km altitude there are differences of up

[Figure]

**Figure 1: (a–e)** and **(g–h)**: Zonal wind differences between seven of the eight radiosonde stations introduced in Fig. 2 in the manuscript and the winds at Mia Padang as a reference. In panel **(f)** the zonal winds at Mia Padang are also given for illustration. Contour lines indicate the ERA-5 zonal mean zonal wind shown in Fig. 1a in the manuscript.

to $-10\,\mathrm{m\,s^{-1}}$ that persist for about three months. Similar negative deviations, albeit weaker, are also seen at Tarawa in the same months and the same altitude range. These deviations may therefore be related to a quasi-stationary wave. (Please note that also in former Figs. 7 and particularly Fig. 8 enhanced QSPW variances are seen in the same months and the same altitude range.) Deviations of similar magnitude and duration, but opposite sign are seen between Tarawa and Mia Padang in early 2019 at 20 to 30 km altitude, and in late 2020 to early 2021 at 18 to 23 km altitude. These differences occur even though the three

mentioned stations are all closer than 1.5° to the equator. This means that observations at single stations can easily be biased and shows the importance of global wind observations from satellite, such as Aeolus, in order to be able to calculate reliable zonal mean zonal winds in the stratosphere.

**(12) L250 : Aren't the intervals at 3 days ? (L188)**

for clarification, we have replaced "7-day intervals" with "7-day averaging periods"

**(13) L265 : Please provide the corresponding period here again.**

the period was added as recommended

**(14) L268 : Where is the SAO shear-related signal found in the figure ?**

The statement about the SAO has been deleted because this effect is seen only in mid 2020 at the very highest altitudes.

**(15) L269 : "drop" : what does this mean ?**

For clarification we have replaced "drop" with "amplitude reduction with altitude"

**(16) L272–274 : Could the uncertainty range of altitude estimates, derived from the temperature and pressure profiles, be calculated/provided ? Would it be as large as ∼1 km at z = 20–25 km ?**

As a measure of uncertainty, for the reanalyses we have calculated at given geopotential altitudes the pressure differences between reanalysis and different radiosonde stations as a reference. These pressure differences were then converted into altitude differences using the local scale height. This has been performed for the dataset of 7-day averages calculated with a time step of three days. The characteristics found for the different radiosonde stations are very similar. Therefore we only show the differences for the Tarawa station where the zonal wind differences in former Fig. 5e–h were most prominent.

[Figure]

**Figure 2:** Differences in geopotential altitude between **(a)** ERA-5, **(b)** JRA-55, and **(c)** MERRA-2 and the radiosondes at Tarawa as a reference. Altitude differences were calculated from pressure differences at given geopotential altitudes using the local scale height. Zonal wind contour lines are from ERA-5. Contour interval is $20\,\mathrm{m\,s^{-1}}$, westward winds are indicated by dashed contour lines, and zero wind is indicated by solid contour lines.

As can be seen from this figure, the altitude differences show a systematic pattern with only small temporal variations. Deviations are typically less than 100 m with JRA-55 showing

the smallest deviations on average (possibly because JRA-55 assimilates IGRA radiosonde data). Given typical magnitudes of QBO-related vertical gradients of the zonal wind of about $10\,\mathrm{m\,s^{-1}\,km^{-1}}$ (perhaps $20\,\mathrm{m\,s^{-1}\,km^{-1}}$ at maximum), this would result in typical biases of 1 to $2\,\mathrm{m\,s^{-1}}$, which is much less than the biases seen for JRA-55. This is some evidence for systematic errors in the zonal momentum budget of JRA-55 being the dominant effect responsible for the wind biases seen in the QBO shear zones. Also for the other reanalyses errors in the zonal momentum budget will contribute to the wind differences seen in the zones of strong wind shear.

See also General Comment 1c by Reviewer # 1. We have included the new figure and the discussion about different altitude scales into a new Appendix B4, which is part of new Appendix B that addresses several potential error sources.

**(17) L279–280 : The shear-related differences found between reanalysis and radiosonde winds (e.g., positive differences along the dashed contour in the stratosphere in 2021–2022, followed by negative differences along the solid contour in 2022; seen more clearly in ERA-5 and JRA-55 than in MERRA-2) are observed at all four stations, thus it is arguable whether the cancellation effect due to zonal averaging could hide these signals. Moreover, these shear-related differences from radiosonde winds are also found in Aeolus data in 2021–2022 along the dashed and solid contours with generally the same signs: positive and negative, respectively (although a bit noisy). The fact that the shear-related differences (from the radiosonde) are rather consistently found in Aeolus and reanalyses can therefore lead to the expectation that differences between Aeolus and reanalysis winds (as in Fig. 1) regarding the shear zone would be less pronounced than the differences from the radiosonde.**

It is correct that the differences between Aeolus and radiosondes partly show a systematic pattern similar to that of the differences between reanalyses and radiosondes, but somewhat weaker. Therefore it is expected that the systematic differences between reanalyses and radiosondes seen in former Figs. 4 and 5 will not show up in Fig. 1 in the manuscript to full extent. Still, there should be some cancellation effects on zonal average.

Therefore we have reworded the text in former lines 279–281 as follows:

"The fact that differences in the shear zones are less pronounced in the zonal averages shown in Fig. 1 could partly be related to cancellation effects that occur in zonal averages. Further, in the shear zones the differences between Aeolus and the radiosondes are often similar to the differences between the reanalyses and the radiosondes, which will reduce the shear zone effect when Aeolus is taken as a reference in Fig. 1. Finally, the altitude coverage of Aeolus at altitudes above 20 km is quite limited, which means that the altitudes where these effects are strongest are not well covered in Fig. 1."

**(18) L327 : "1 cycle/day" : How is this determined ?**

For clarification, we have added the following sentence:

"This frequency limit results from the twice-daily asynoptic satellite sampling by combining ascending and descending parts of the satellite orbit that are measured at different solar local times (e.g., Salby, 1982)."

**(19) L337–339 : That the phase shift is mentioned here leads readers to deduce the whole procedures to obtain the two spectra (while there could exist different ways) which have been omitted in the text. If the authors would mention the phase shift in the text, an explanation for the procedures before/after the shifting should also be supplemented. Or it may also be okay if the phase shift — a specific way to obtain the two spectra — is not mentioned at all, while still referring to the reference for the details.**

As suggested, the sentence mentioning the phase shift has been removed in the revised manuscript.

**(20) L363–364 : Is there any reason that the different frequency ranges are used between reanalyses and Aeolus ? Would it be more consistent if the frequencies higher than 1 cycle/day are not used in reanalyses too?**

We are limiting our study to frequencies $<0.5\,\mathrm{cycles\,day^{-1}}$, i.e., well below the Nyquist frequencies of all datasets. Therefore, the higher Nyquist frequencies of the reanalyses of $2\,\mathrm{cycles\,day^{-1}}$ compared to $\sim 1\,\mathrm{cycle\,day^{-1}}$ for Aeolus should not be relevant.

**(21) L417 : What would be a possible reason that the reanalyses could not capture this? (ENSO-related phenomena must be in large scales in space and time.)**

ENSO as a possible explanation was given just as a speculation, and it was clearly mentioned that further investigation was needed.
Still, there is some justification for our speculation:
It is correct that ENSO-related phenomena are large scale in space and time. However, the circulation shifts of the Pacific Walker circulation occur in a region (the Pacific) where wind observations are relatively sparse! This means that the reanalyses are less well constrained in this region, which is indicated by the relatively poor performance of the reanalyses (even of ERA-5) at the location of Tarawa where the circulation shift becomes most obvious.
Therefore adding ENSO as a possible explanation and clearly marking this as a speculation seems to be justified.

We have reworded our statement in former l.417, added some explanation, and emphasized more clearly that our explanation is very speculative:

"Possibly, the reanalyses are not fully capturing this circulation shift because in the Pacific region, where this circulation shift happens, there are only few observations from radiosondes that can be assimilated in the reanalyses. Consequently, reanalyses might be less reliable in the Pacific region. Still, our data record is far too short to be able to reliably attribute the larger deviations in Fig. 8b, d, and f to ENSO-related phenomena, and our suggested explanation remains very speculative."

**(22) L464–465 : Regarding the underrepresentation of MRGWs in JRA-55, Kim et al. (2019, doi:10.5194/acp-19-10027-2019) could also be added here (in which the meridional winds are used for MRGWs).**

Reference has been added as recommended.

**(23) Fig. 10 : It is interesting to observe the MRGW peak above the easterly, around January 2022 (z ∼ 30 km). Would there be no statement to mention about this ?**

The following explanation has been added in the revised manuscript:

"Further, it should be mentioned that enhanced variances at high altitudes are not necessarily due to equatorial MRGWs, but could also be caused by extratropical Rossby waves propagating equatorward from the winter hemisphere. In particular, enhanced variances around 30 km in January 2022 are suspicious because this variance peak is observed above a QBO westward phase where MRGWs propagating from below could have been filtered out."

**(24) L471 : "are similar (about 10–15 m2 s–2)"**

changed as recommended

**(25) L484 : "n=0 inertia gravity waves" (with no parenthesis)**

changed as recommended

**(26) L546 : The La Nina condition may be a possible factor for the less performance seen in 2021–2022. Could you please revise the sentence to be a bit less conclusive ? (The 5-year record is too short to generalize this finding.)**

We have revised the sentence to clearly mark our explanation as very speculative:

"Further, deviations for the quasi-stationary waves seem to be somewhat larger in the years 2021 and 2022. We speculated that this might be related to the circulation shift by La Niña conditions in these years. However, our data record is far too short to allow more than a vague speculation, and more detailed investigations would be needed."

**(27) Appendix A1 : "Rossby-wave-like", "gravity-wave-like" : I would suggest using a less ambiguous term.**

In the original work by Matsuno (1966) the term "Rossby wave type" was used for the equatorial Rossby wave solutions. Therefore we will use this term in the revised manuscript, referring to Matsuno (1966). For the other wave modes (Kelvin waves, IGWs, and MRGWs) we will use the term "other wave modes" because this class of wave modes includes the MRGWs which are a mixture of the "Rossby wave type" and the "inertio-gravity wave type" modes that were introduced by Matsuno (1966). Still, it is correct that these "other wave modes" have at least some properties of inertia-gravity waves.

**(28) L582 : Could you please explain this a bit more ? (I could not understand it.)**

For clarification, we have added the following explanation:

"This means that wind amplitudes and temperature amplitudes of Rossby waves maximize for different conditions. If at a given altitude the geopotential perturbation due to a Rossby wave maximizes, the corresponding strong horizontal geopotential gradients will likely also lead to a maximum of the wind perturbation. Different from this, at the altitude of the maximum geopotential perturbation, the vertical gradient of the geopotential perturbation will be zero. Consequently, the temperature perturbation will be small."

**(29) L591 : The stability change around the tropopause also leads to the decrease in the vertical wavelength which then could result in the change in the zonal wind amplitudes. Would it be justified to consider the constant wind amplitude here ?**

For clarification, we have added the following explanation in the revised manuscript:

Of course, the assumption of a constant wind amplitude is somewhat arbitrary and was just made to facilitate the explanation of the temperature amplitude jump at the tropopause. As can be seen from former Fig. 9, the zonal wind variances of Kelvin waves reduce by a factor of about 4 between the upper troposphere and the lower stratosphere, which means a reduction of the Kelvin wave zonal wind amplitudes by a factor of about two. Accordingly, a reduction in temperature amplitudes by the same factor would be expected assuming a constant buoyancy frequency. However, as can be seen from Fig. A1d-f, we observe even the opposite effect: temperature amplitudes increase by a factor of about two between the upper troposphere and the lower stratosphere, which underlines the importance of the effect of changes in the buoyancy frequency at the tropopause.

**Reply to the Technical Comments by Reviewer # 2:**

**(1) L32 : "sensitive" : (to what ?) not clear what this means**

"very sensitive" has been deleted

**(2) L40 : "These" → "The"**

done

**(3) L68 : "observation" : repetitive**

second occurrence has been removed

**(4) L84 : "tropical" : repetitive**

first occurrence has been removed

**(5) L88 : "mean zonal wind" (to distinguish from what is investigated in Sect. 4)**

changed as recommended

**(6) L115 : "and neglecting" → "neglecting" ?**

changed as recommended

**(7) L161 : "pressures below" → "pressures less than"**

done

**(8) Fig. 1 caption : (1) The period "2018-2022" is applied not only for the mean but also for the standard deviation. Please rephrase the sentence. (2) Currently what the standard deviation is of is not specified (which must be of the difference). Please rephrase it. (3) Black solid and dashed lines : What are the corresponding values ? (0 and –20 m/s ?)**

Caption of Fig. 1 has been revised.

**(9) Fig. 1 : Could the late months in 2022 in ERA-5 now be filled ?**

The ERA-5 dataset has been extended until end of 2022. Further, the zonal wind contour lines are now from ERA-5 as was recommended in Specific Comment (10).

**(10) L207 : comma → period**

done

**(11) L288–289 : Please re-write "This . . . stations" to, e.g., "The magnitude of this pattern differs among the stations."**

Thank you very much for this improved formulation! Changed as recommended.

**(12) L293 : "differences" → "errors" ? (such advection or wave forcing cannot be derived from sonde data, and thus the comparison/difference cannot be made.)**

replaced as recommended

**(13) L296 : "shear, deficiencies" (comma)**

done

**(14)** Fig. 6 caption : (1) Is it actually the "squared spectral amplitudes", not the spectral variances ? (2) Would it be clearer if n = 1 (sym.) and n = 2 (antisym.) were indicated for Rossby waves, to distinguish from n > 2 modes ? (Or, could the term "equatorial" Rossby waves already imply those ?)

(1) Squared spectral amplitudes is correct. A factor of 0.5 has to be applied after integrating in the spectral domain over the squared spectral amplitudes for obtaining variances as shown later in Sect. 4.2. This was already stated at the beginning of Sect. 4.2.1 where an example was given for the symmetric QSPW1. This example showed that the QSPW1 amplitude determined from the spectra is in good agreement with the QSPW1 amplitude that can be estimated from the radiosondes.

(2) Strictly speaking, also higher modes of equatorial Rossby waves could occur in the atmosphere — for example, n=3 symmetric, or n=4 antisymmetric modes, or even higher-order modes. These higher-order modes, however, should be of smaller amplitude. Higher-order modes differ from lower-order modes by their meridional structure, but cannot be distinguished from the lower-order modes in the latitudinally averaged spectra presented in former Fig. 6: the equatorial Rossby wave contributions in former Fig. 6 are the sum of all symmetric and antisymmetric equatorial Rossby waves, respectively, irrespective of their order. This means that using the general terms "symmetric equatorial Rossby waves" and "antisymmetric equatorial Rossby waves" is correct as is.
However, we now note in the figure caption that the equatorial Rossby wave dispersion lines are plotted for the n=1 and n=2 modes only, and a short explanation is given in the text of the revised manuscript.

**(15)** L358 : "In addition" → "Here" ; "... avoided in the calculation of the background"

changed as suggested

**(16)** L373–374 : I could not understand this sentence.

This sentence has been rewritten by adding some more explanation. It now reads:

"As the variance of a sine wave averaged over a full period is 0.5 times the squared amplitude of the wave, the peak variances seen in Fig. 7a, c, e, and g correspond to a 15°S–15°N and 31-day average amplitude of square-root two times the variance, i.e., the square-root of $300\,\mathrm{m^2\,s^{-2}}$, which is about $17\,\mathrm{m\,s^{-1}}$."

**(17)** L428–433 : Too many "Again" (4 out of 6 sentences in the paragraph begin with this.)

Three out of four "again" have been replaced, or deleted.

**(18)** L496 : "initiatives" : plural or singular ?

As both cited references are from the same initiative we are using singular now.

**(19)** L580 : "z" in the equation may also need to be defined (log-pressure altitude).

Definition of $z$ has been added.

**(20)** L585 : "gravity-wave-like Kelvin" → "Kelvin"

changed as suggested

**(21)** Please remove commas ahead of 'and' in, e.g., L20, 29, 345.

done

**(22) Please remove commas ahead of 'or' in, e.g., L41, 45, 312.**

done

**References**

Bushell, A. C., Anstey, J. A., Butchart, N., Kawatani, Y., Osprey, S. M., Richter, J. H., Serva, F., Braesicke, P., Cagnazzo, C., Chen, C.–C., Chun, H.–Y., Garcia, R. R., Gray, L. J., Hamilton, K., Kerzenmacher, T., Kim, Y.–H., Lott, F., McLandress, C., Naoe, H., Scinocca, J., Smith, A. K., Stockdale, T. N., Versick, S., Watanabe, S., Yoshida, K., and Yukimoto, S.: Evaluation of the Quasi-Biennial Oscillation in global climate models for the SPARC QBO-initiative, Q. J. Roy. Meteor. Soc., 148, 1459–1489, https://doi.org/10.1002/qj.3765, 2022.

Butchart, N.: The Brewer-Dobson circulation, Rev. Geophys., 52, 157–184, doi:10.1002/2013RG000448, 2014.

Matsuno, T.: Quasi-geostrophic motions in the equatorial area, J. Meteorol. Soc. Jpn., 44, 25–43, 1966.

Mote, P. W., Rosenlof, K. H., McIntyre, M. E., Carr, E. S., Gille, J. C., Holton, J. R., Kinnersley, J. S., Pumphrey, H. C., Russell III, J. M., and Waters, J. W.: An atmospheric tape recorder: The imprint of tropical tropopause temperatures on stratospheric water vapor, J. Geophys. Res., 101, 3989–4006, doi:10.1029/95JD03422, 1996.

Pumphrey, H. C., Boone, C., Walker, K. A., Bernath, P., and Livesey, N. J.: Tropical tape recorder observed in HCN, Geophys. Res. Lett., 35, L05801, doi:10.1029/2007GL032137, 2008.

Salby, M. L.: Sampling theory for asynoptic satellite observations, Part I: Space-time spectra, resolution, and aliasing, J. Atmos. Sci., 39, 2577–2600, 1982.